# PolyMOF nanoparticles constructed from intrinsically microporous polymer ligand towards scalable composite membranes for $CO_2$ separation

Tae Hoon Lee[1,2], Byung Kwan Lee[1], Seung Yeon Yoo[1], Hyunhee Lee[2], Wan-Ni Wu[2], Zachary P. Smith [2] & Ho Bum Park [1] ✉

Integrating different modification strategies into a single step to achieve the desired properties of metal−organic frameworks (MOFs) has been very synthetically challenging, especially in developing advanced MOF/polymer mixed matrix membranes (MMMs). Herein, we report a polymer−MOF (polyMOF) system constructed from a carboxylated polymer with intrinsic microporosity (cPIM-1) ligand. This intrinsically microporous ligand could coordinate with metals, leading to ~100 nm-sized polyMOF nanoparticles. Compared to control MOFs, these polyMOFs exhibit enhanced ultramicroporosity for efficient molecular sieving, and they have better dispersion properties in casting solutions to prepare MMMs. Ultimately, integrating coordination chemistries through the cPIM-1 and polymer-based functionality into porous materials results in polyMOF/PIM-1 MMMs that display excellent $CO_2$ separation performance (surpassing the $CO_2/N_2$ and $CO_2/CH_4$ upper bounds). In addition to exploring the physicochemical and transport properties of this polyMOF system, scalability has been demonstrated by converting the developed MMM material into large-area (400 $cm^2$) thin-film nanocomposite (TFN) membranes.

$CO_2$ capture from existing fossil fuel power plants plays a crucial role in mitigating global average atmospheric $CO_2$ concentration for achieving carbon neutrality to ensure a sustainable future[1]. Although amine absorption processes are a leading technology for post-combustion capture of $CO_2$, this method is highly energy-intensive, consuming ~30% of the power produced by the plant, and thus does not meet the target $CO_2$ capture cost estimated by the U.S. Department of Energy (DOE)[2]. Membrane separation has emerged as an economical alternative due to its operational convenience, small footprint, excellent scalability, and potentially high energy efficiency[3–5]. However, it has been challenging further to improve the $CO_2$ separation efficiency of membrane processes to compete with other mature technologies since conventional polymeric membrane

materials are governed by an inherent trade-off relationship between permeability and selectivity[3].

Hybridization of the mechanically robust polymeric matrix and molecularly selective inorganic fillers to fabricate mixed matrix membranes (MMMs) is a facile and efficient strategy to improve the separation efficiency of pure polymers by integrating the advantages of both phases[6]. Among the numerous material candidates, metal−organic frameworks (MOFs) have been explored as a promising filler due to their remarkably tunable characteristics, such as pore size, porosity, topologies, dimensions, and chemical functionalities which allow researchers to customize them to prepare MMMs depending on the target separation applications[7–9]. Despite tremendous research efforts over two decades, industrial deployment of MOF-based MMMs

[1]Department of Energy Engineering, Hanyang University, Seoul 04763, Republic of Korea. [2]Department of Chemical Engineering, Massachusetts Institute of Technology, Cambridge, MA 02139, USA. ✉e-mail: badtzhb@hanyang.ac.kr

has yet to be realized due to several remaining challenges, including filler–matrix incompatibility[10–13], particle agglomeration[14,15], insufficient operational stability (e.g., plasticization and physical aging)[16,17], and limited scalability[8,18].

To overcome these issues, post-synthetic modifications of MOFs have been widely investigated, including the introduction of additional functional groups[19,20], ligand (or metal) exchange[21,22], and surface oligomer/polymer coating[12,13,15,23–25]. For example, polymethyl methacrylate (PMMA)-functionalized UiO-66 nanoparticles showed excellent colloidal stability in casting solutions, which led to enhanced particle dispersion and interactions within the polymer matrix[20]. Li et al. functionalized UiO-66-NH$_2$ nanoparticles by covalent grafting with polyimide brushes with the same molecular structure as the polymer matrix, resulting in a strong brush-brush interaction based on the rule of 'like dissolves like'[12,15]. Reducing the particle sizes of MOFs by modulated synthesis is another strategy to improve the separation performance of MMMs by increasing the interfacial area between the polymer matrix and MOF fillers[14,16,26,27]. However, integrating these strategies into a single step is very synthetically challenging when trying to achieve the desired properties of MOFs[8,28]. Specific challenges include the following: (i) post-modifications by surface coating generally reduce the accessible pore volume of MOFs[12,20,24,25], (ii) modulated synthesis could lead to particle size variations and structural defects in MOFs, which cause uncertainty in MMMs[29–31], and (iii) reducing MOF particle size below 100 nm accelerates self-agglomeration induced by the Ostwald ripening effect[14,32]. Hence, a unified synthetic approach of multifunctional MOFs is necessary to provide more opportunities for developing advanced MMMs.

In addition to the reported polymer–MOF hybrids fabricated by coating, grafting, in situ polymerization, and MMM approaches, polymer–metal–organic frameworks (polyMOFs) consisting of amorphous and linear polymer ligands coordinated with metal ions have recently gained much attention as a unique class of hybrid materials that combines the features of both polymer and MOFs[28,33–35]. Cohen and co-workers first reported the concept of polyMOFs by coordinating Zn$^{2+}$ with a poly(benzenedicarboxylic acid) (pBDC) ligand (Fig. 1)[34]. Follow-up studies of polyMOFs have explored the structural effects of polymer ligands[36–39], hierarchical structure and porosity[40,41], isoreticular chemistry[42], and the use of block copolymer ligands[41,43]. Johnson et al. reported that the particle size and colloidal stability of polyMOF-5 nanoparticles could be simultaneously controlled using a multivalent polyMOF ligand. For adsorbent applications, IRMOF-1 type polyMOFs showed high CO$_2$ sorption but very low N$_2$ sorption by a kinetic sieving effect coupled with their exceptional water stability[35].

These examples demonstrate the excellent potential of polyMOF concepts for tailoring the physicochemical properties of existing MOFs for efficient CO$_2$ separation. However, reported polyMOF chemistries and their applications are still rare, and polyMOFs suffer from a substantial reduction in surface area (more than half of the loss compared to parent MOFs) since the pores are occupied by the non-porous polymer chains[34,40,42].

To this end, we propose a multifunctional polyMOF system constructed from a microporous polymer (i.e., polymers of intrinsic microporosity, PIM) ligand, which both modulates the characteristics of polyMOF nanoparticles by a one-step synthesis and provides the framework with angstrom-scale microporosity for molecular sieving (Fig. 1). The coordination reactions between the PIM ligand and metal ions were investigated by detailed microscopic, spectroscopic, and thermal analyses, confirming the successful synthesis of PIM-based polyMOF nanoparticles. Compared to the control MOFs fabricated by organic ligand, these polyMOFs exhibited particle size reduction, enhanced ultramicroporosity (3–4 Å), and better colloidal stability, making them favorable to prepare high-performance polyMOF/PIM-1 MMMs for CO$_2$ separation. Furthermore, the scalability of developed MMM material was demonstrated by preparing a large-area thin-film membrane.

## Results

### Synthesis of polyMOF nanoparticles containing cPIM-1 ligand

In this study, carboxylated PIM-1 (cPIM-1) was selected as a potential PIM ligand given that this PIM contains a benzenedicarboxylic acid (BDC) unit, which can coordinate with the metal ions to form polyMOFs analogous to its organic ligand counterpart (i.e., BDC). The synthesis and characterization of cPIM-1 is described in Supplementary Note 1 and Supplementary Figs. 1–5. Importantly, cPIM-1 was soluble in polar aprotic solvents such as N, N-dimethylformamide (DMF), which is a common solvent for MOF synthesis[44].

Next, polyMOFs containing cPIM-1 were fabricated via a mixed-ligand approach to obtain crystalline structures, and their synthetic conditions are listed in Supplementary Table 1[28,38]. Initially, we explored the UiO-66 type polyMOFs coordinated with Zr clusters. These were designated as polyUiO-66(x:y), where x:y is the molar ratio between BDC and cPIM-1 used for their synthesis (Fig. 2a). For the pure organic ligand case (i.e., x:y = 1:0), UiO-66 was used to represent the control MOF. Photo images of the polyUiO-66 samples displayed homogeneous powders, and the color of the powder becomes darker following the color of cPIM-1 (dark brown) and increasing the cPIM-1 concentration (Supplementary Fig. 6). A transmission electron

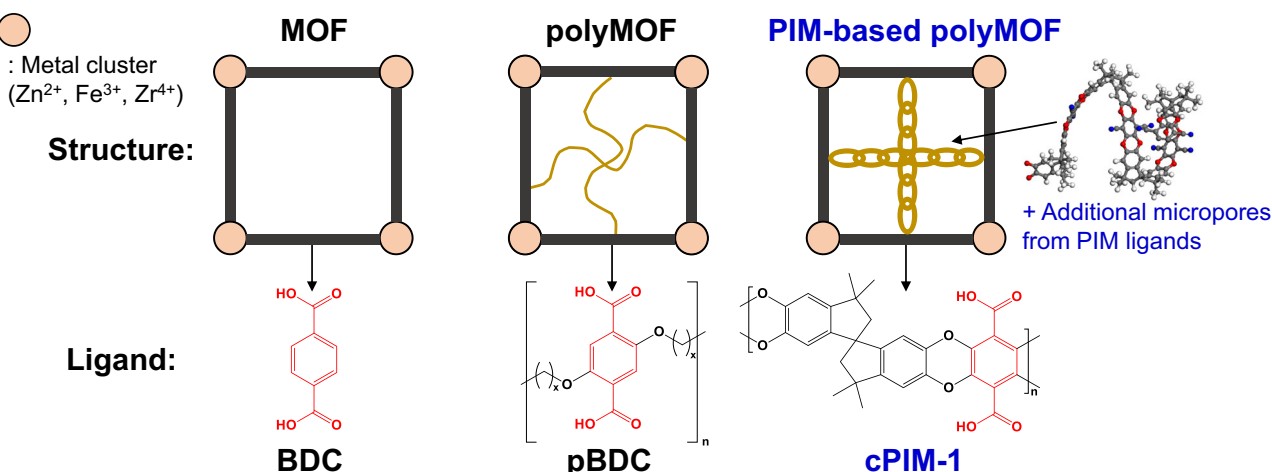

**Fig. 1 | Schematic illustration.** Metal–organic framework (MOF), polymer–MOF (polyMOF), and polymers of intrinsic microporosity (PIM)-based polyMOF system constructed from different types of ligand containing benzenedicarboxylic acid (BDC, highlighted by red color) unit. Note: pBDC = poly(benzenedicarboxylic acid) and cPIM-1 = carboxylated PIM-1, respectively.

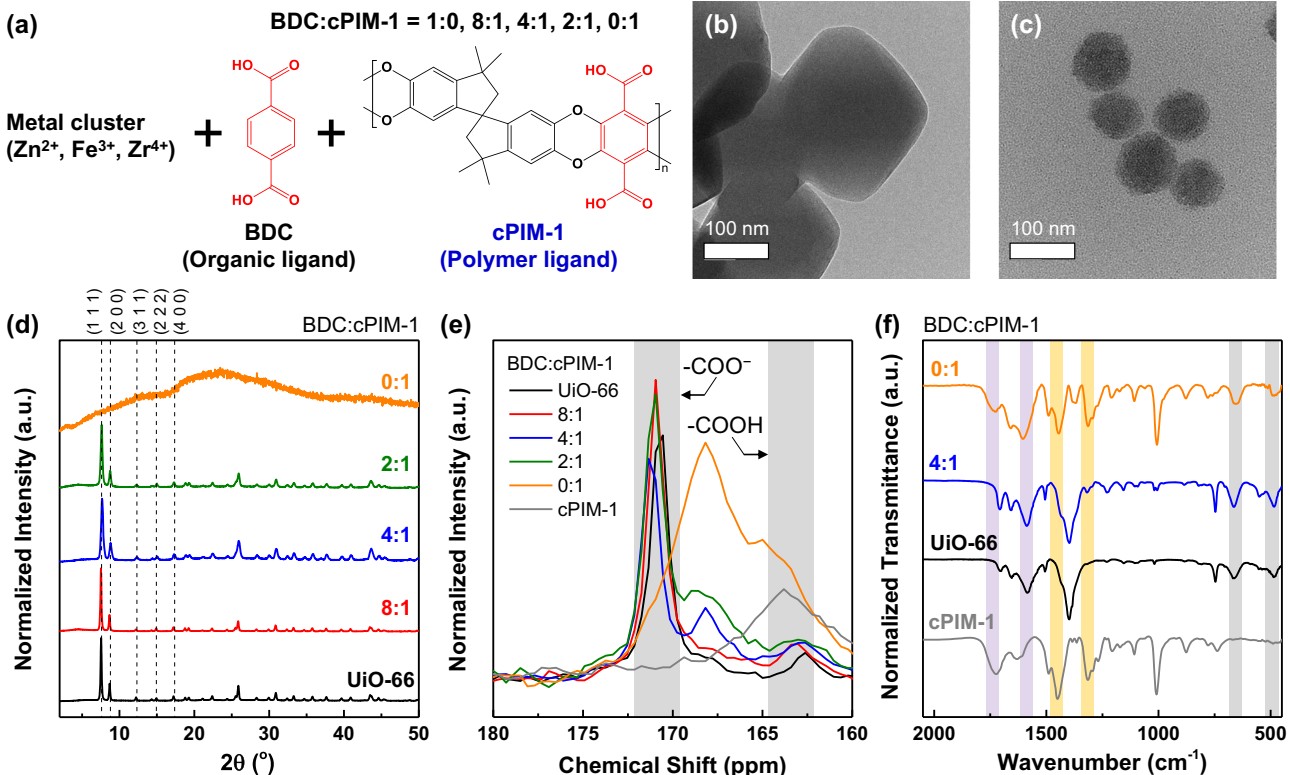

**Fig. 2 | Characterization of polyMOFs. a** Synthesis of polyMOF nanoparticles containing cPIM-1 ligand. Transmission electron microscopy (TEM) images of **b** UiO-66 and **c** polyUiO-66(4:1) nanoparticles. **d** Powder x-ray diffraction (XRD) patterns, **e** solid-state $^{13}C$ nuclear magnetic resonance (NMR) spectra, and **f** Fourier-transform infrared spectroscopy (FT-IR) spectra of polyUiO-66 samples depending on BDC:cPIM-1 ratio used for their synthesis. Note: yellow highlights: methylene ($CH_2$) stretching/bending vibration modes and the C–O stretching mode, purple highlights: carboxylate bands for cPIM-1 and the red-shifted peaks for polyUiO-66 samples, and gray highlights: Zr oxo clusters in polyUiO-66 samples.

microscopy (TEM) image of UiO-66 showed ~200 nm-sized octahedral nanoparticles, while that of polyUiO-66(4:1) exhibited spherical nanoparticles with smaller particles less than 100 nm (Fig. 2b, c). High-resolution TEM images also reveal that the morphology of the polyUiO-66 nanoparticles becomes rougher by increasing the cPIM-1 concentration (Supplementary Fig. 7). These measurements were supported by scanning electron microscopy (SEM) observations (Supplementary Fig. 8a, b). The particle size reduction in polyUiO-66(4:1) may be attributed to the higher viscosity (1.80 cP at 20 °C) of the precursor solution for polyUiO-66(4:1) synthesis compared to that for UiO-66 synthesis (1.24 cP at 20 °C), which decreases the diffusivity of reactants and thus retards the growth of the nanoparticles[45].

Powder x-ray diffraction (XRD) pattern of polyUiO-66(0:1) showed a completely amorphous nature (Fig. 2d) as its SEM image displayed interconnected, uneven particle morphologies (Supplementary Fig. 8c). A recent study combining density functional theory and solid-state nuclear magnetic resonance (ssNMR) reported that if the polymer ligand does not fit the molecular restraints of the MOF lattice, structural distortions and defects can occur in the resulting polyMOFs[39]. For polyUiO-66, extending the length of alkyl spacers in the pBDC linker resulted in amorphous structures, which suggests there is an upper limit on the size of the repeating unit needed to generate polyMOFs[40]. cPIM-1 possesses a bulky, highly contorted spirobisindane unit and a ladder-like rigid chain structure, which may not be accommodated in the pores of UiO-66 due to the high mechanical constraints[6,39,46]. In addition, the molecular weight of the repeating unit of cPIM-1 (~498 g/mol) is similar to that of the extended pBDC linker (x = 10 in Fig. 1, ~476 g/mol), which failed to afford crystalline polyUiO-66[40]. These support our result that using only the cPIM-1 ligand led to an amorphous material. Even so, the crystallinity of

UiO-66 was maintained for BDC:cPIM-1 ratio up to 2:1, suggesting that crystalline polyMOFs could be synthesized by judiciously controlling the ratio between organic ligand and cPIM-1[28,38].

Since cPIM-1 was mixed with BDC to form crystalline materials, it raises the question of whether the polymer ligand has truly been integrated into the MOF lattice, or if it is merely coating the surface of UiO-66 crystals that were formed separately[38]. First, a TEM image of polyUiO-66(4:1) confirmed that the particles were homogenous without any noticeable phase separation throughout their cross-sections (Fig. 2c), which implies the cPIM-1 ligands were not solely coated onto the surface of nanoparticles. Also, the chemical structures of polyUiO-66 samples were explored by $^{13}C$ magic angle spinning (MAS) ssNMR spectroscopy. It is well documented that the coordination of linkers in polyUiO-66 is linked to the deprotonation of the carboxylate groups in the BDC unit, which can be identified by a shift in the $^{13}C$ NMR spectrum from approximately 165 to 170 ppm[39]. As the cPIM-1 concentration for the polyUiO-66 synthesis increased, the peak for uncoordinated groups (−COOH, at ~163 ppm)[44] became more intense, while that for coordinated groups (−COO⁻, at ~171 ppm)[39] became broader and moved toward the peak for the −COOH group (Fig. 2e). These results indicate that the incorporation of cPIM-1 into the UiO-66 lattice induces structural defects in the resulting polyUiO-66[39], which provides further evidence that the surface binding of cPIM-1 is not the sole mechanism to form the polyUiO-66. Of note, in contrast to the cPIM-1 that showed only a −COOH peak, the NMR spectra of polyUiO-66(0:1) exhibited both −COOH and −COO⁻ portions, which suggests that there are specific coordinations between Zr metal and cPIM-1 ligands in polyUiO-66(0:1) despite the absence of crystallinity.

The coordination chemistry in polyUiO-66 can be attributed to two possible interactions: one is ionic crosslinking by Zr ions[47,48] and

the other is coordination between Zr oxo clusters and cPIM-1 (Supplementary Fig. 9)[49,50]. To judge which one is the governing mechanism, Fourier-transform infrared (FT-IR) spectra of cPIM-1, UiO-66, polyUiO-66 (4:1), and polyUiO-66(0:1) were compared (Fig. 2f). The yellow highlighted peaks at 1450 and 1310 $cm^{-1}$ correspond to the methylene ($CH_2$) stretching/bending vibration modes and the C−O stretching mode, respectively, which were intensified by increasing the cPIM-1 concentration[51]. The purple highlighted peaks indicate that the carboxylate bands at 1724 $cm^{-1}$ for cPIM-1 were significantly red-shifted to 1583 $cm^{-1}$ for polyUiO-66 samples, as observed in control UiO-66. This is ascribed to the coordination between the incorporated Zr metal and the polymer ligand[38]. The strongest uncoordinated carboxylate IR peak observed for polyUiO-66(0:1) agrees with the NMR analyses. Notably, the peaks at 483 and 661 $cm^{-1}$ (gray highlights) confirmed the presence of Zr oxo clusters in all polyUiO-66 samples[50]. This implies that the polyUiO-66(0:1) was also formed in a manner similar to the crystalline UiO-66, which makes it classified as an amorphous MOF[52]. Additionally, the residual mass at 800 °C obtained from thermogravimetric analysis (TGA) curves under air purge (18.3 wt.%) is far above that typically observed in other studies (<5 wt.%) on ionic crosslinking of cPIM-1 (Supplementary Fig. 10 and Supplementary Table 2), even for polyUiO-66(0:1)[38,47,48]. Taken together, the results demonstrate that the cPIM-1 ligand could interpenetrate through the MOF lattice by coordinating with metal oxo clusters rather than the simple ionic crosslinking, thus forming crystalline polyMOFs, which illustrates that we can modulate the properties of MOFs.

cPIM-1-based coordinations were extended to other metal ions such as $Zn^{2+}$ and $Fe^{3+}$, and the relevant polyMOFs were designated as polyMOF-5(x:y) and polyMIL-101(x:y), respectively, where x:y is the molar ratio between BDC and cPIM-1 used for their synthesis. Again, both polyMOFs exhibited a uniform powder that darkened in color as the cPIM-1 to BDC ratio increased (Supplementary Figs. 11 and 12), and the SEM images revealed that particle sizes of polyMOF-5(4:1) and polyMIL-101(4:1) were smaller than those of control MOF-5 and MIL-101, respectively (Supplementary Figs. 13 and 14). When cPIM-1 was used in the absence of BDC (i.e., x:y = 0:1), amorphous materials were observed by both SEM images and powder XRD pattern, while the crystallinity of the control MOFs remained up to a BDC:cPIM-1 ratio of 4:1 (Supplementary Fig. 15). FT-IR spectra of polyMOF-5 and polyMIL-101 samples revealed the presence of metal oxo clusters ($Zn_4O$ for MOF-5[53] and $Fe_3O$ for MIL-101[54], respectively) as well as cPIM-1 ligand based on the red-shifts in the −COOH group peak of cPIM-1 (Supplementary Fig. 16). The residual mass analyses by TGA (Supplementary Fig. 17) also confirmed the high concentration of Zn or Fe metal clusters in the corresponding polyMOFs (Supplementary Tables 3 and 4). These results are consistent with the above observations in polyUiO-66 cases, which proves the successful synthesis of polyMOFs from various metal sources and the generality of our concept on the use of cPIM-1 as a polymer ligand.

## Enhanced ultramicroporosity in polyUiO-66 nanoparticles

In the following sections, we focused on the polyUiO-66 materials given their particle sizes down to ~100 nm, which is within a typically explored range for MOF-based MMM studies and may fit into the preparation of high-flux MMM thin films[14,26,27]. $N_2$ sorption isotherms at 77 K showed a lower sorption capacity of the polyUiO-66 samples compared to that of UiO-66 and the essentially non-porous nature of the amorphous polyUiO-66(0:1) (Fig. 3a). According to the non-local density functional theory (NLDFT) model, the $N_2$-based pore size distribution of polyUiO-66 represents smaller micropores (<20 Å) with a narrower distribution as the cPIM-1 concentration increases (Supplementary Fig. 18a). To obtain a more precise evaluation of the angstrom-scale pores, additional analysis of the gas adsorption and pore size distribution was performed using $CO_2$ as a probe gas given its smaller kinetic diameter (3.30 Å) compared to $N_2$ (3.64 Å)[55].

Interestingly, the $CO_2$ sorption capacity at 273 K for polyUiO-66 was enhanced by increasing the BDC:cPIM-1 ratio from 1:0 to 4:1, which is markedly opposed to the $N_2$ sorption results (Fig. 3b).

The Brunauer−Emmett−Teller (BET) surface areas determined from both $N_2$ and $CO_2$ sorption isotherms are summarized in Fig. 3c. Compared to the control UiO-66, the $CO_2$-based surface area of polyUiO-66(4:1) was slightly increased from 770 to 818 $m^2\,g^{-1}$, while the $N_2$-based surface area of polyUiO-66 samples was reduced from 1492 to 1306 $m^2\,g^{-1}$ as the cPIM-1 concentration increased. The improvement in ultramicroporosity (<7 Å)[56] for PIM-based polyUiO-66 materials is very unique, as evidenced by the increased $CO_2$ surface area. Loss in porosity or surface area has been a challenging topic for polyMOF materials due to pore filling by extra polymer chains or ligand units, as observed for the $N_2$ surface area measurements of polyUiO-66 that mainly probe large microporosity (7−20 Å)[42,55,56]. From $CO_2$ sorption isotherms at 273 and 298 K (Supplementary Fig. 19a), only marginal variations were found in the isosteric heat of $CO_2$ adsorption (25−28 kJ/mol) between UiO-66 and polyUiO-66 materials (Supplementary Fig. 19b). This allows us to exclude the potential effects of favorable interactions between polyUiO-66 and the probe $CO_2$ molecules upon incorporation of the cPIM-1 ligand[57]. Hence, the structural changes caused by incorporated cPIM-1 chains would govern the resulting microporosity (or ultramicroporosity) of polyUiO-66.

To gain further insight, pore size distributions from $CO_2$ sorption at 273 K were calculated from the NLDFT model. The majority of ultramicropores in UiO-66 and polyUiO-66 exist around 5−6 Å, which is consistent with the reported pore size of UiO-66 (6 Å) (Supplementary Fig. 18b)[50]. Notably, the pore volumes at 3−4 Å, where pure cPIM-1 is also observed, intensified by increasing the cPIM-1 concentration (Fig. 3d). Therefore, we speculated that although the large micropores were mostly occupied by incorporating cPIM-1 ligands, the intrinsic microporosity of cPIM-1 may still offer additional ultramicropores in the resulting polyMOF system, and these are responsible for the enhanced ultramicroporosity of polyUiO-66.

The potential application of polyUiO-66 as $CO_2$ adsorbents was evaluated by ideal $CO_2/N_2$ adsorption selectivity at 298 K (Supplementary Figs. 19c, d). In addition to the higher $CO_2$ uptake at 1 bar for polyUiO-66(4:1) (1.79 mmol $g^{-1}$) than that of UiO-66 (1.69 mmol $g^{-1}$), the $CO_2/N_2$ selectivity of polyUiO-66(4:1) was improved compared to control UiO-66 by 27% at 0.1 bar and 14% at 1 bar. The obtained $CO_2/N_2$ selectivity of polyUiO-66(4:1) is highest among the UiO-66-based adsorbents with a similar level of $CO_2$ uptake (Supplementary Table 5). The excellent $CO_2/N_2$ selectivity of polyUiO-66(4:1) is mainly attributed to the presence of ultramicropores, especially in the 3−4 Å range, which may contribute to the more pronounced molecular sieving effect that allows the diffusion of smaller $CO_2$ molecules while retarding that of larger $N_2$ molecules[35]. Thus, polyUiO-66(4:1) was chosen as a representative polyMOF for fabricating MOF/polymer MMMs.

## Enhanced colloidal stability of polyUiO-66 nanoparticles

Dispersion of MOFs in the casting solution is a key factor that governs the polymer−MOF interfacial compatibility and thus the separation ability of MMMs[8]. However, MOF nanoparticles generally suffer from self-agglomeration due to their tendency to interact with each other[28]. To visually confirm the colloidal stability, photo images of UiO-66 and polyUiO-66(4:1) dispersions in several common solvents (concentration = 0.1 mg $mL^{-1}$) were taken after 7 days. Significant precipitation of UiO-66 nanoparticles was observed by the naked eye in all tested solvents (Fig. 4a). On the other hand, polyUiO-66(4:1) showed a stable dispersion in polar aprotic solvents such as DMF and tetrahydrofuran (THF), which are good solvents for cPIM-1 as well. Acetone swells the cPIM-1, and some precipitation of polyUiO-66(4:1) particles was seen. Complete sedimentation occurred in chloroform ($CHCl_3$),

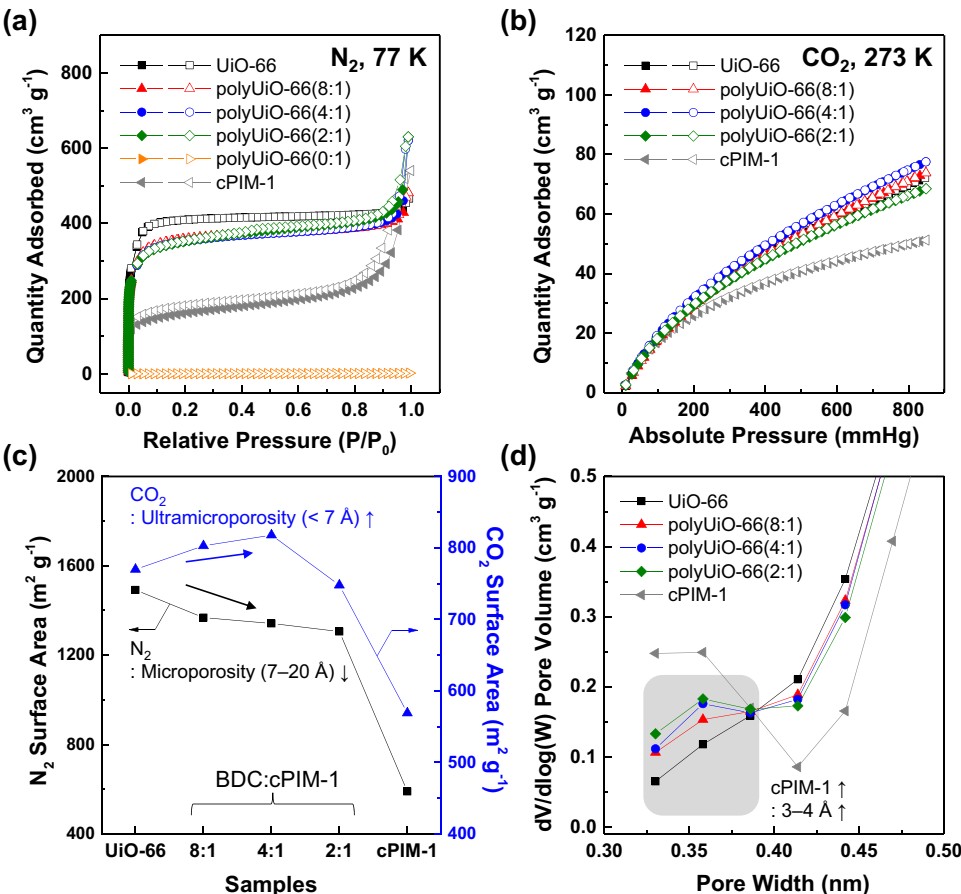

**Fig. 3 | Microporosity analyses. a** $N_2$ sorption isotherms at 77 K and **b** $CO_2$ sorption isotherms at 273 K of polyUiO-66 samples (filled symbols: adsorption and unfilled symbols: desorption). **c** Brunauer–Emmett–Teller (BET) surface area of polyUiO-66 samples calculated from $N_2$ sorption isotherms and $CO_2$ sorption isotherms, respectively. **d** Non-local density functional theory (NLDFT) pore size distributions

of polyUiO-66 nanoparticles calculated from the $CO_2$ adsorption isotherm at 273 K. Gray squares highlight the pore size range between 3 to 4 Å. Note that polyUiO-66(0:1) was excluded for $CO_2$ sorption studies due to its essentially non-porous nature.

which is a poor solvent for cPIM-1. These results indicate that the dispersibility of polyUiO-66(4:1) nanoparticles follows the solubility of cPIM-1, which may be attributed to the existence of unoccupied cPIM-1 ligands on their external surface that enhance colloidal stability, especially in good solvents for cPIM-1[28]. In the same manner, a coating of polyimide (6FDA-Durene) oligomer onto MOFs has been shown to facilitate the formation of a uniform dispersion of oligomer-MOF hybrid particles in the polymer solution since both the polymer and MOF surface have the same functionality and solubility[25].

The excellent colloidal stability of polyUiO-66(4:1) was also proved by dynamic light scattering (DLS) method, which confirmed average diameters of ~100 nm in both DMF and THF, which agreed well with the TEM and SEM analyses (Fig. 4b, c). In contrast, the control UiO-66 displayed very large particle size distributions (more than 1 μm) in both solvents, which is well above their original sizes (~200 nm). Eventually, the high concentration dispersion (8 mg L⁻¹ in THF) of UiO-66 was entirely sedimented within 24 h, while that of polyUiO-66(4:1) remained stable and homogeneous (Supplementary Fig. 20), which is favorable for actual MMM casting.

**Preparation and characterization of polyUiO-66/PIM-1 MMMs**
Overall, our PIM-based one-step synthetic approach provides the resulting polyMOFs with multiple benefits such as particle size reduction, enhanced ultramicroporosity, and excellent colloidal stability, which are desirable characteristics to fabricate defect-free and high-performance MMMs. PIM-1 was selected as a polymer matrix

given its structural similarity with the cPIM-1 ligand, which may improve the interfacial compatibility and $CO_2$ separation ability[6]. The UiO-66/PIM-1 and polyUiO-66(4:1)/PIM-1 MMMs were designated as Uxx/PIM-1 and pUxx/PIM-1, respectively, where xx indicates the loading amount of UiO-66 or polyUiO-66(4:1) (5, 10, and 20 wt.%). Photo images of U20/PIM-1 and pU20/PIM-1 MMMs display clear differences in optical transparency and top/bottom views (Fig. 5a, b), which emphasize the macroscopic homogeneity of the pU20/PIM-1 film. This is mainly ascribed to the stable dispersion of polyUiO-66(4:1) nanoparticles in the casting solution, while that of UiO-66 was so insufficient that most of the UiO-66 nanoparticles aggregated and sedimented during solvent evaporation for film formation.

PIM-1 is a highly rigid polymer that is notorious for causing significant filler−matrix interfacial voids at both macro and molecular scales as described in experimental and computational studies[58,59]. Likewise, cross-sectional SEM analyses of U20/PIM-1 revealed the presence of significant agglomerates as well as interfacial gaps between filler and matrix, which are known to deteriorate the separation performance of MOF/polymer MMMs (Fig. 5c, d)[8]. In contrast, polyUiO-66(4:1) nanoparticles are uniformly distributed throughout the PIM-1 matrix, and no noticeable defects or interfacial microvoids were detected, even for high-magnification images (Fig. 5e, f). Additional characterizations on MMMs were performed to support the microscopic observations. The characteristic XRD peaks of UiO-66 are evident in those of the MMMs without any peak shifts (Fig. 5g), indicating that the crystalline structure of incorporated

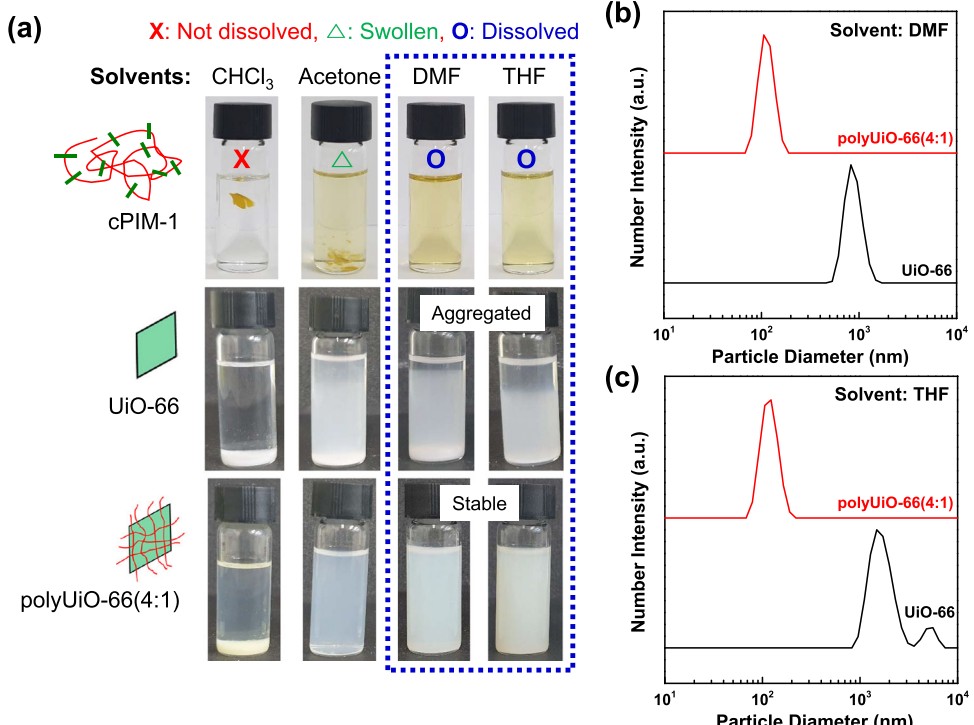

**Fig. 4 | Colloidal stability. a** Photo images of dissolution of cPIM-1 and dispersion of UiO-66 and polyUiO-66(4:1) nanoparticles in different solvents. The images were taken 7 days after dissolving (or dispersing) materials into each solvent via ultrasonication for 1 h. Particle size distributions of UiO-66 and polyUiO-66(4:1) nanoparticles dispersed in **b** dimethylformamide (DMF) and **c** tetrahydrofuran (THF) determined by dynamic light scattering (DLS) measurements (concentration = 0.1 mg mL$^{-1}$).

fillers was maintained inside the matrix, and partial infiltration of the PIM-1 chains can be neglected[30,31]. pU20/PIM-1 exhibited greater improvements in mechanical properties when compared to PIM-1 and U20/PIM-1, as evidenced by its increased hardness and reduced modulus (Fig. 5h) as determined from the load-displacement curves of nanoindentation tests (Supplementary Fig. 21). Furthermore, $N_2$ sorption analyses of PIM-1 and MMM dense films revealed that the BET surface areas followed the order of pU20/PIM-1 (747 m$^2$ g$^{-1}$) > PIM-1 (689 m$^2$ g$^{-1}$) > U20/PIM-1 (638 m$^2$ g$^{-1}$) (Fig. 5i). Considering the higher BET surface area of MOF fillers than that of PIM-1 matrix, the incorporation of polyUiO-66(4:1) nanoparticles into PIM-1 matrix is responsible for the improved surface area of pU20/PIM-1[60]. On the other hand, the reduced surface area of U20/PIM-1 is attributed to the significant particle agglomeration and interfacial microvoids, which may diminish the accessible pores of incorporated UiO-66 fillers. Note that the lower surface area of PIM-1 film compared to that of powder form (861 m$^2$ g$^{-1}$) is ascribed to the phase inversion process during PIM-1 synthesis, which leads to an irregular morphology in the powder[61]. Taken together, these results are consistent with the SEM analyses, indicating that incorporating polyUiO-66(4:1) into PIM-1 leads to better compatibility between the filler and matrix at their interfaces compared to the use of control UiO-66. The improved filler–matrix adhesion in pU20/PIM-1 can be explained by the existence of uncoordinated cPIM-1 ligands on the polyUiO-66(4:1) nanoparticle surface as evidenced by the dispersion stability tests that support the 'like dissolves like' principle, which is a widely accepted approach for improving the dispersibility of MOFs in a polymer matrix via surface modification[12,15,25].

## CO$_2$ separation performance of polyUiO-66/PIM-1 MMMs
Pure-gas CO$_2$ separation performances of prepared MMMs were evaluated for CO$_2$/N$_2$ and CO$_2$/CH$_4$ pairs depending on the filler concentration (Fig. 6a, b). Despite its high CO$_2$ permeability, a pure PIM-1

membrane has been limited for CO$_2$ separation applications due to its low CO$_2$/gas selectivity and stability issues[6]. Incorporating UiO-66 nanoparticles into PIM-1 matrix only improved CO$_2$ permeability, whereas CO$_2$/N$_2$ and CO$_2$/CH$_4$ selectivities were significantly reduced with increasing UiO-66 concentration possibly due to the defects from the significant particle agglomerations. In contrast, polyUiO-66(4:1)/PIM-1 MMMs exhibited a significant increase in both CO$_2$ permeability (from 2822 to 9659 Barrer, 1 Barrer = 10$^{-10}$ cm$^3$ (STP) cm cm$^{-2}$ s$^{-1}$ cmHg$^{-1}$) and CO$_2$/N$_2$ selectivity (from 14.4 to 21.5) by increasing the filler concentration up to 20 wt.%, and a similar enhancement effect was found in the CO$_2$/CH$_4$ pair. The improved CO$_2$ separation performances were also observed in the MMMs containing polyUiO-66(4:1) filler with different polymer matrices (Supplementary Figs. 22 and 23). This emphasizes the versatility of the PIM-based polyMOF filler design. Ultimately, CO$_2$ separation performances of pU20/PIM-1 membrane far surpassed the 2008 Robeson upper bound, which are comparable to the state-of-the-art membrane materials for CO$_2$ separation, such as polyethylene oxide (PEO) derivatives, thermally rearranged (TR) polymers, PIMs, and MMMs[62].

To better understand the transport mechanism, the CO$_2$ and N$_2$ transport properties of each PIM-1, U20/PIM-1, and pU20/PIM-1 membranes were examined within the framework of the solution-diffusion model, which defines the permeability coefficient ($P_i$) of a penetrant i through a membrane as the product of its diffusion coefficient ($D_i$) and solubility coefficient ($S_i$), expressed as $P_i = D_i \times S_i$[63] (Supplementary Fig. 24). More than one order of magnitude higher $D_i$ was found for PIM-1 and MMMs when compared with traditional glassy polymers due to the enormous intrinsic micropores in the PIM-1 matrix[64]. U20/PIM-1 showed an 86%-increased $D_{CO_2}$ compared to those of PIM-1, while a significant drop in CO$_2$/N$_2$ diffusivity selectivity ($D_{CO_2}/D_{N_2}$, 2.5 to 2.0) was observed with almost identical $S_{CO_2}$ and CO$_2$/N$_2$ solubility selectivity ($S_{CO_2}/S_{N_2}$). The reduced diffusivity selectivity can be explained by the observed particle agglomerations and interfacial

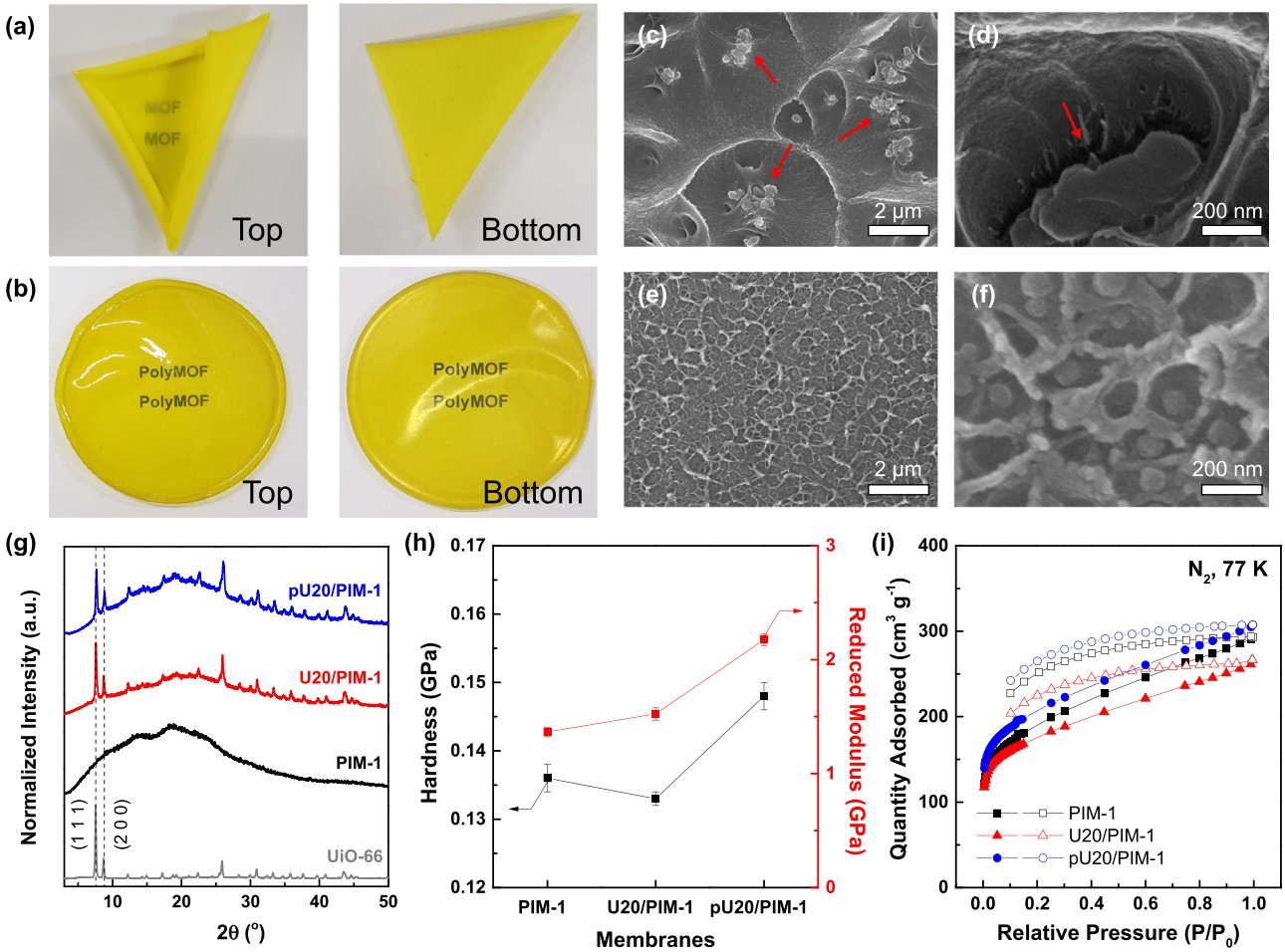

**Fig. 5 | Characterization of MMMs.** Photo images of **a** U20/PIM-1 and **b** pU20/PIM-1 membranes. Cross-sectional scanning electron microscopy (SEM) images of **c**, **d** U20/PIM-1 and **e**, **f** pU20/PIM-1 membranes. Red arrows indicate the significant agglomerates and interfacial voids in U20/PIM-1 mixed matrix membrane (MMM). **g** XRD spectra, **h** mechanical properties calculated from the nanoindentations tests (error bars represent standard deviations from 5 different analyses), and **i** $N_2$ sorption isotherms at 77 K (filled symbols: adsorption and unfilled symbols: desorption).

defects in U20/PIM-1[8]. On the other hand, pU20/PIM-1 displayed a more pronounced improvement in both $D_{CO_2}$ by 147% and $S_{CO_2}$ by 38%, which led to both enhanced diffusivity selectivity (2.5–2.9) and solubility selectivity (5.7–7.4). These features are ascribed to the effective molecular sieving by ultramicroporous polyUiO-66(4:1) filler, uniform particle dispersion, and improved filler–matrix compatibility, which are responsible for the simultaneous increase in $CO_2$ permeability and $CO_2/N_2$ selectivity of pU20/PIM-1[65,66].

Physical aging, i.e., the time-dependent relaxation of non-equilibrium free volume elements, is a major concern in high-free-volume glassy polymers such as PIM-1[6]. After 365 days of aging, PIM-1, U20/PIM-1, and pU20/PIM-1 membranes all displayed a significant decrease in $CO_2$ permeability (unfilled stars in Fig. 6a, b). Nevertheless, the $CO_2$ separation abilities of the aged pU20/PIM-1 were still above the upper bound and were accompanied by a moderate increase in $CO_2/N_2$ (or $CO_2/CH_4$) selectivity due to the densification of PIM-1 matrix over time[46].

Selectivity loss by penetrant-induced plasticization is another critical concern in $CO_2$ separation membranes because of the high condensability of $CO_2$ molecules[62]. For glassy polymers, the plasticization pressure refers to the threshold at which the gas permeability begins to increase with increasing pressure. That is, the higher the plasticization pressure, the higher the plasticization resistance of membranes[9,12]. Notably, pU20/PIM-1 showed the highest $CO_2$ plasticization pressure (~36 bar), while U20/PIM-1 exhibited a lower

plasticization pressure (~16 bar) than that of PIM-1 (~26 bar) (Fig. 6c). The results are consistent with the excellent $CO_2$ separation abilities of pU20/PIM-1 as well as strong interfacial interactions between the polyUiO-66(4:1) and PIM-1 matrix that matches the molecular structure of cPIM-1 ligand, which can significantly restrict the mobility of the matrix polymer and thereby enhance the plasticization resistance of MMM[8,9,12]. Ultimately, $CO_2/N_2$ mixture permeation tests (50:50 mol.%) revealed a lower decrease in separation performances from pure-gas to mixed-gas for pU20/PIM-1 when compared to PIM-1 and U20/PIM-1 (Fig. 6d). Together with the ultrahigh $CO_2$ permeability (~9000 Barrer) in a mixed-gas condition, the $CO_2/N_2$ mixed-gas selectivity of 20.0 for pU20/PIM-1 meets the required $CO_2/N_2$ selectivity for post-combustion $CO_2$ capture (>20)[2].

### Scale-up demonstration of polyUiO-66/PIM-1 MMMs

To deploy the developed materials in actual $CO_2$ separation processes, the freestanding, bulk films explored for fundamental transport studies should be converted into a thin-film composite (TFC) membrane configuration, consisting of a thin selective layer (<3 μm) that offers lower mass transport resistance and a porous support layer that provides adequate mechanical stability[18,46]. Adding nano-sized fillers (i.e., MOFs) into the selective layer results in thin-film nanocomposite (TFN) membranes. To demonstrate the feasibility of scaling up the high-performance pU20/PIM-1 MMM, its TFN membrane was fabricated using a scalable bar-coating method[18]. We could

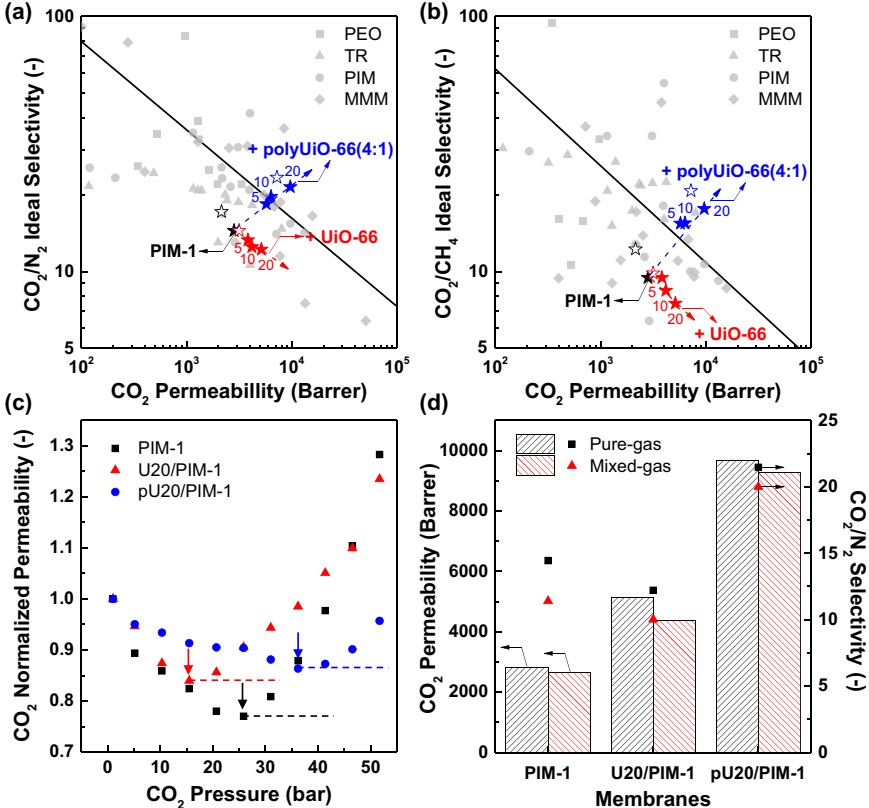

**Fig. 6 | $CO_2$ separation performances.** Pure-gas separation performances of PIM-1, UiO-66/PIM-1, and polyUiO-66(4:1)/PIM-1 membranes for **a** $CO_2/N_2$ and **b** $CO_2/CH_4$ compared with literature data[62]. Black solid lines indicate the proposed $CO_2/N_2$ and $CO_2/CH_4$ pure-gas upper bounds for polymeric membranes, respectively. Numbers next to the filled symbols indicate the concentration (wt.%) of each filler. Unfilled symbols indicate the gas separation performances of 365-day-aged PIM-1, U20/PIM-1, and pU20/PIM-1 membranes, respectively. **c** $CO_2$ plasticization study (arrows: plasticization pressure) and **d** $CO_2/N_2$ mixed-gas (50:50 mol%) separation performance of PIM-1, UiO-66/PIM-1, and polyUiO-66(4:1)/PIM-1 membranes.

uniformly enlarge the TFN membrane from a small area ($3 \times 3$ cm$^2$) to a large area ($20 \times 20$ cm$^2$), and no significant pinholes or aggregated particles were visually detected (Fig. 7a). A $3 \times 3$-cm$^2$-sized TFC membrane consisting of ~2.6 μm-thick pure PIM-1 as a selective layer was also prepared (Fig. 7b) as a control sample, while the TFN membrane possesses pU20/PIM-1 selective layer (~2.7 μm) consisting of homogeneously distributed polyUiO-66(4:1) nanoparticles (Fig. 7c). Again, this is attributed to the good dispersion of the polyUiO-66(4:1) filler, which maximizes the solution processability of TFN membranes. The TFN membranes successfully exhibited more than two times higher $CO_2$ permeance and improved $CO_2/N_2$ selectivity regardless of the membrane area when compared to those of the TFC membrane as observed in the bulk film studies (Fig. 7d). In contrast, ~3.3 μm-thick U20/PIM-1 TFC membrane showed a similar $CO_2$ permeance (~2400 GPU, 1 GPU = $10^{-6}$ cm$^3$ (STP) cm$^{-2}$ s$^{-1}$ cmHg$^{-1}$) with that of PIM-1 TFC membrane while accompanying a significantly reduced $CO_2/N_2$ selectivity (10.5), which is attributed to the severe agglomeration of UiO-66 nanoparticles (Supplementary Fig. 25). Of note, the $CO_2$ permeance of TFC and TFN membranes (~2000 and ~4800 GPU, respectively) significantly differed from the predicted permeances from the resistance-in-series model (~1100 and ~3600 GPU, respectively)[3], which may be attributed to the potential penetration of the casting solution into the porous substrate and the microstructural changes in PIM-1 matrix during the thin-film formation as recently reported[46].

The achieved $CO_2/N_2$ separation performance of the large-area ($20 \times 20$ cm$^2$) pU20/PIM-1 TFN membrane was compared with the literature data, and it was located within the target area required for the membrane-based post-combustion $CO_2$ capture (Fig. 7e)[2]. To the best

of our knowledge, this TFN membrane possesses the largest area among the $CO_2$ separation membranes based on pure polymer or MMM and it displayed a similar level of either $CO_2$ permeance or $CO_2/N_2$ selectivity. These findings highlight the potential of the pU20/PIM-1 TFN membrane in industrial $CO_2/N_2$ separation processes, enabled by multifunctional polyMOF design based on the PIM ligand. Although significant aging-induced permeance reduction was found in both TFC and TFN membranes (Supplementary Fig. 26), the extent was much less for the TFN membrane, possibly due to the favorable filler–matrix interactions[8]. The aging behavior can be potentially addressed by nonsolvent-based rejuvenation[67] or we could also exploit the aging-induced selectivity enhancement[46]. For example, the 14-days-aged PIM-1 TFC and pU20/PIM-1 TFN membranes could be rejuvenated by simply soaking them into methanol, which was also repeatable (Supplementary Fig. 26). Further optimization of coating conditions may accelerate its commercialization.

## Discussion

In conclusion, we demonstrated a multifunctional polyMOF system based on the cPIM-1 ligand and its potential application for membrane-based $CO_2$ separation. This one-step synthetic approach is facile, generalizable to different metals, and effective to fine-tune the physicochemical properties of MOFs (e.g., particle size and coordination chemistry). In particular, the cPIM-1 ligand offers enhanced ultra-microporosity in the resulting polyMOF, overcoming one of the critical drawbacks of polyMOFs, which is that their porosity and surface area are drastically reduced from those of parent MOFs. In addition, the cPIM-1 ligand enables better colloidal stability coupled with enhanced filler–matrix interfacial compatibility to fabricate defect-free MMMs

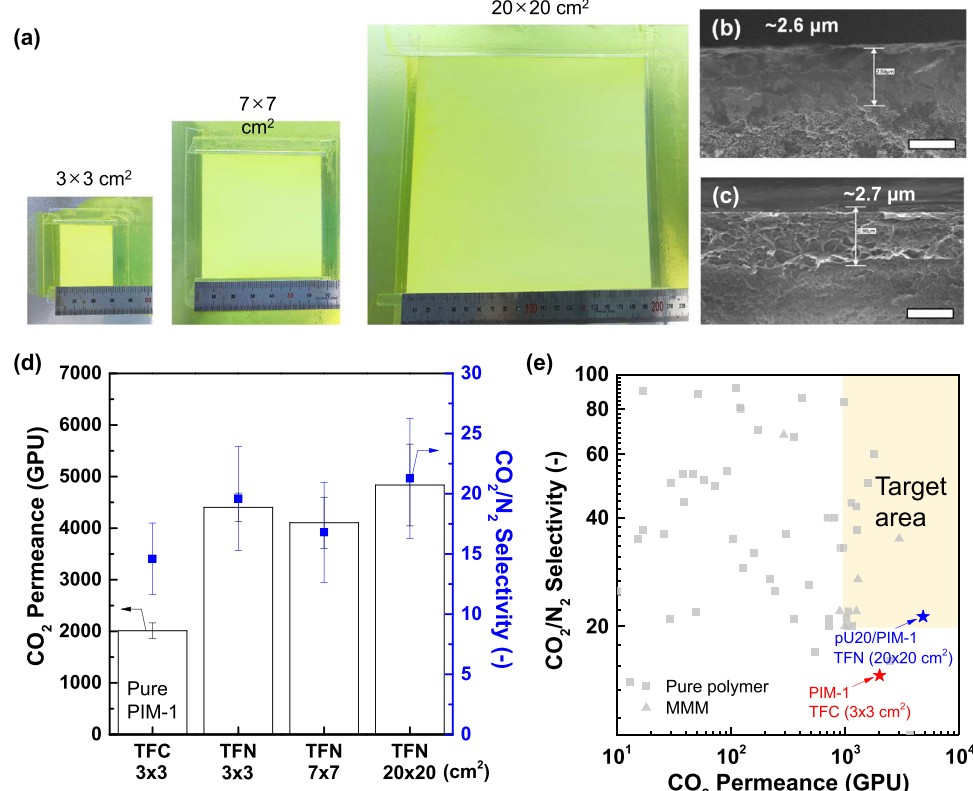

**Fig. 7 | Scalability demonstration. a** Photo images of as-prepared pU20/PIM-1 thin-film nanocomposite (TFN) membranes depending on membrane area. Cross-sectional SEM images of **b** PIM-1 thin-film composite (TFC) membrane ($3 \times 3$ cm²) and **c** pU20/PIM-1 TFN membranes ($20 \times 20$ cm²) (scale bar = 2 μm). **d** $CO_2/N_2$ separation performance of TFC and TFN membranes depending on membrane area

tests (error bars represent standard deviations from 4 different samples). **e** Comparison of $CO_2/N_2$ separation performances of TFC and TFN membranes prepared in this study with literature data[70]. The yellow square denotes the desired performance for the membrane-based post-combustion $CO_2$ capture[2].

based on the 'like dissolves like' rule. The polyMOF-incorporated MMM displayed both excellent $CO_2$ separation abilities surpassing the limitation of pure polymers and superior plasticization resistance compared to those of pure polymer and control MOF-based MMMs. The scalability of developed MMM material was also examined by the successful fabrication of a defect-free and large-area TFN membrane. We anticipate the proposed strategy will overcome the existing challenges in both MOF and polyMOF materials, leading to advanced MOF/polymer MMMs for energy-efficient molecular separations.

## Methods
### Materials
Zirconium(IV) chloride ($ZrCl_4$, 99.5%), zinc nitrate hexahydrate ($Zn(NO_3)_2 \cdot 6H_2O$, 98%), iron(III) chloride hexahydrate ($FeCl_3 \cdot 6H_2O$, 97%), 1,4-benzenedicarboxylic acid (BDC, 98%), potassium carbonate ($K_2CO_3$, 99%), N,N-dimethylformamide (DMF, 99.8%, anhydrous), tetrahydrofuran (THF, 99%), chloroform ($CHCl_3$, 99.5%), dichloromethane ($CH_2Cl_2$, 99.5%), acetic acid (99%, glacial), sulfuric acid ($H_2SO_4$, 98%), methanol (MeOH, 99.8%), and acetone (99.5%) were purchased from Sigma Aldrich (USA). Formic acid (85%) was purchased from Daejung Chemicals & Metals (South Korea). Tetrafluoroterephthalonitrile (TFTPN, 99%) was purchased from Matrix Scientific (USA) and purified by vacuum sublimation at 145 °C. 5,5',6,6'-Tetrahydroxy-3,3,3',3'-tetramethyl-1,1'-spirobisindane (TTSBI, 97%) was purchased from Alfa Aesar and recrystallized in MeOH/$CH_2Cl_2$ solution before use[68]. Porous polyacrylonitrile (PAN) membranes with a molecular weight cut-off (MWCO) of 10 kDa (type: UF 010104) were purchased from Solsep BV (Netherlands). 6FDA-DAM and 6FDA-DAM:DABA(3:2) polymers were purchased from Akron Polymer Systems (USA). Matrimid® 5218 (Matrimid) polymer was purchased from

Alfa Aesar (USA). Gases for permeations tests were purchased from Air Korea (South Korea). Deionized (DI) water was obtained from a Milli-Q water purification system (Millipore, USA).

### Synthesis of PIM-1
PIM-1 was synthesized via a modified method according to a previous report[68]. TTSBI (10.213 g, 30 mmol), TFTPN (6.003 g, 30 mmol), and $K_2CO_3$ (8.292 g, 60 mmol) were dissolved in DMF (200 mL) in a round-bottom flask attached with a water condenser under nitrogen purge and were stirred for 20 min at room temperature. Then, the reaction flask was heated to 70 °C and stirred for 72 h. After that, THF (400 mL) was poured into the flask and stirred for 30 min to separate the solution into a supernatant solution containing low-molecular weight polymers with cyclic chains and a precipitated polymer phase. After filtration, the precipitated phase was dissolved into THF and separated into a supernatant solution containing high-molecular weight PIM-1 with residual $K_2CO_3$ and a precipitated cross-linked PIM-1 phase. After filtration, the supernatant was washed with methanol to remove residual monomers and redissolved into THF. The redissolved solution was washed with DI water to remove residual $K_2CO_3$. Finally, a bright yellow PIM-1 powder was collected using vacuum filtration and dried under vacuum at 100 °C for 24 h.

### Synthesis of carboxylated PIM-1 (cPIM-1)
cPIM-1 was synthesized via a solid-state acid hydrolysis reaction based on a previous report[44]. PIM-1 (0.3 g), DI water (18 mL), acetic acid (6 mL), and sulfuric acid (18 mL) were added sequentially to a round-bottom flask attached with a water condenser. The reaction flask was heated up to 150 °C and stirred for 48 h. After that, DI water (500 mL) was added to the solution, and the brown powder was collected by

filtration. To eliminate residual reagents, the dark brown powder was refluxed with DI water (200 mL) and 3 drops of sulfuric acid at 95 °C for 12 h. It was then filtered and dried in a vacuum oven at 130 °C for 24 h.

## Synthesis of MOFs and polyMOFs

In a typical reaction, each metal source (ZrCl₄, Zn(NO₃)₂·6H₂O, and FeCl₃·6H₂O), organic ligand (BDC), and polymer ligand (cPIM-1) were dissolved in DMF in a glass vial (30 mL) at room temperature depending on the molar ratios of reagents (Supplementary Table 1). The reaction was performed in a pre-heated oven (120 °C for UiO-66 series, 100 °C for MOF-5 series, and 110 °C for MIL-101(Fe) series) for 48 h. The resulting powder (or brittle film) was sequentially washed with DMF (×2), MeOH (×1), and CHCl₃ (×2). The precipitate was collected and dried in a vacuum oven at 120 °C for 24 h. Here, the poly-MOFs containing the cPIM-1 ligand were designated as polyUiO-66(x:y), polyMOF-5(x:y), and polyMIL-101(x:y) where x:y is the molar ratio between BDC and cPIM-1 for their synthesis. Note that formic acid was added for the polyUiO-66 synthesis as it has been reported that modulators are critical for UiO type polyMOF synthesis[40,42]. The yield of polyMOF nanoparticles was approximately 60–70% by weight.

## Preparation of mixed matrix membranes (MMMs)

A predetermined amount of UiO-66 (U) or polyUiO-66(4:1) (pU) nanoparticles were dispersed in THF by ultrasonication for 60 min. After that, PIM powder was dissolved in the MOF/THF solution to form a 4 wt.% solution. The solution was poured into a Teflon dish, covered by a glass plate, and slowly evaporated for more than 24 h at room temperature. The resulting films were dried in a vacuum oven for 24 h at 40 °C before characterization. The UiO-66/PIM-1 and polyUiO-66(4:1)/PIM-1 MMMs were designated as Uxx/PIM-1 and pUxx/PIM-1, respectively, where xx indicates the loading amount of UiO-66 or polyUiO-66(4:1) (5, 10, and 20 wt.%). Pure PIM-1 membrane was prepared by the same procedure except for adding MOFs into the casting solution. The thickness of bulk membranes was adjusted to 80 ± 10 μm by changing the volume of the casting solution.

## Preparation of thin-film nanocomposite (TFN) membranes

For the preparation of TFN membranes, UiO-66 or polyUiO-66(4:1) nanoparticles (20 wt.% in polymer) were dispersed in THF by ultrasonication, and PIM-1 powder (4 wt.% in THF) was subsequently dissolved in the MOF/THF solution. The coating solution was cast onto the PAN support membrane using an automated bar-coater (HAN-TECH, South Korea) at a coating speed of 60 mm/s and room temperature. The prepared TFN membranes were dried in a vacuum oven at room temperature for 24 h before characterization. Thin-film composite (TFC) membranes (pure PIM-1 as a selective layer) were prepared by the same procedure except for adding MOFs into the casting solution.

## Characterization

A field-emission scanning electron microscope (FE-SEM, JSM-700F, JEOL, Japan) and a transmission electron microscope (TEM, JEM 2100 F, JEOL, Japan) were used to observe the morphologies of MOF nanoparticles and membranes. The crystalline structure of samples was investigated using an X-ray diffractometer (XRD, Miniflex 600, Rigaku, Japan) with focused monochromatized Cu Kα radiation (λ = 1.5418 Å) at a scan rate of 10°/min. Fourier-transform infrared (FT-IR) spectra were analyzed using a Nicolet 6700 spectrometer (Thermo Fisher Scientific, USA) to investigate the functional groups of MOFs. Solid-state $^{13}$C magic angle spinning (MAS) nuclear magnetic resonance (NMR) experiments were performed using a Bruker Avance III HD 400 MHz spectrometer (Bruker, Germany). The fine powder samples were packed in a 4 mm zirconia rotor. For each NMR spectrum, 512 scans were recorded at a 10 kHz spinning rate with a pulse length of 1 μs and a recycle delay of 5 s. The thermal stability of MOFs was examined using a thermal gravimetric analyzer (TGA, Q500, TA Instruments, USA) under continuous flow of N₂ or air. The surface area and pore size distribution of polymers, MOFs, and membranes were evaluated by N₂ sorption for micropores at 77 K and CO₂ sorption was conducted to measure ultramicropores at 273 K using a physisorption analyzer (3Flex, Micromeritics, USA). Samples were degassed at 120 °C for 24 h before the sorption measurements. The pore size distribution was obtained by the non-local Density Functional Theory (NLDFT) model assuming a carbon-slit pore geometry using the software package provided by the supplier. Mechanical properties of prepared membranes were evaluated using a nanoindenter (TI-950, Bruker, USA) equipped with a Berkovich probe tip. The probe tip was aligned perpendicular to the membrane surfaces. The load-displacement curves were obtained with the maximum load of 4 mN to calculate the hardness and the reduced modulus. The tests were repeated at least 5 times for each sample and the average values were reported[55]. The particle size distribution of MOF nanoparticles was examined using a dynamic light scattering (DLS) instrument (ELSZ-1000, Otsuka Electronics, Japan). Precursor solution viscosity was measured using a viscometer (DV3T, Brookfield, USA) at 20 °C.

## Calculation of cPIM-1 concentration in polyMOFs

The concentration of cPIM-1 in each polyMOF was evaluated using a residual mass of each component at 800 °C obtained from TGA curves as follows[69]:

$$W_{MOF} + W_{cPIM} = \qquad (1)$$

$$R_{MOF}W_{MOF} + R_{cPIM}W_{cPIM} = R_{polyMOF} \qquad (2)$$

where $W_{MOF}$ and $W_{cPIM}$ are the weight fraction of parental MOF and cPIM-1 in polyMOF and $R_{MOF}$, $R_{cPIM}$, and $R_{polyMOF}$ are the residual mass (wt.%) of parental MOF, cPIM-1, and polyMOF at 800 °C (Supplementary Tables 3–5).

## Gas permeation tests of membranes

The gas permeability coefficient (P) of the freestanding membranes was measured using the constant-volume/variable-pressure method at 35 °C and a pressure difference of 2 bar[10]. Before measurements, both the feed and permeate sides were evacuated by a high-vacuum pump to a pressure of less than 10⁻⁶ Torr. After the feed gas was introduced to the membrane, the permeability coefficient was evaluated from the pressure increase as a function of time at steady-state according to the following equation:

$$P = \frac{VT_0 l}{p_0 T \Delta p A} \left( \frac{dp}{dt} \right) \qquad (3)$$

where V (cm³) is the volume of permeate side, l (cm) is the membrane thickness, Δp (cmHg) is the pressure variation between the feed and permeate side, A (1.13 cm²) is the membrane area, T (K) is the temperature, $T_0$, $p_0$ are the standard temperature and pressure, respectively, and (dp/dt) is the rate of pressure increase at steady-state.

The ideal selectivity was calculated as the ratio of the permeability coefficient of the two single-component gases:

$$\text{Ideal selectivity} = \frac{P_A}{P_B} \qquad (4)$$

The pure-gas transport properties of the TFC and TFN membranes were evaluated by custom-built constant-pressure/variable-volume equipment at 35 °C and a pressure difference of 2 bar[46]. Gas permeance of a penetrant i ($J_i$, unit: GPU, 1 GPU = 10⁻⁶ cm³ (STP) cm⁻² s⁻¹ cmHg⁻¹ = 3.35 × 10⁻¹⁰ mol m⁻² s⁻¹ Pa⁻¹) was calculated as

follows:

$$J_i = Q_i/(\Delta p \cdot A) \qquad (5)$$

where $Q_i$ is the gas flux of penetrant i, $\Delta p$ is the pressure difference, and A is the effective area of the membrane (1.13 cm$^2$).

The $CO_2/N_2$ mixed-gas permeation properties of membranes also were tested using the constant-pressure/variable-volume method using a cross-flow cell with an $CO_2/N_2$ gas mixture (50:50, mol%) at feed gas pressure of 2 bar and a temperature of 35 °C. The stage-cut (permeate to feed flow rate) was controlled to less than 1% by adjusting the retentate flow to mitigate the concentration polarization effect. The composition of the permeate flow was analyzed by a calibrated Agilent 6890 N gas chromatograph equipped with a thermal conductivity detector (TCD).

### Gas sorption measurements of membranes

The gas sorption capacity of the membranes was examined by the pressure decay method at 35 °C using a custom-made dual-volume sorption apparatus equipped with a transducer[10]. At least 0.5 g of membrane coupons were filled into the sample chamber, which was evacuated for 12 h to eliminate any guest molecules. Thereafter, the feed gas (2 bar) was injected into the chamber, and the pressure decay resulting from the sorption of gas molecules in the membranes was recorded. The amount of pressure reduction after reaching steady state was used to determine the gas concentration sorbed in the samples (C, cm$^3$ (STP) cm$^{-3}$ (sample)). The solubility coefficient (S, cm$^3$ (STP) cm$^{-3}$(sample) atm$^{-1}$) at the equilibrium fugacity f (atm) was obtained by the following equation: S = C/f.

## Data availability

All data shown in main text and supplementary information are available from the corresponding author upon request.

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

## Acknowledgements
This work was supported by a National Research Foundation of Korea (NRF) grant funded by the Korean government (MSIT) (No. 2022R1A5A1032539).

## Author contributions
All authors contributed to the scientific discussion and manuscript preparation. Prof. H.B.P. contributed to conceptualization, supervision, and funding acquisition. T.H.L. led the experimental design, data curation, and writing of the original manuscript. B.K.L. and S.Y.Y. contributed to the gas transport studies. H.L. contributed to the $N_2$ and $CO_2$ adsorption studies. W.-N.W. contributed to the synthesis of microporous polymers. Prof. Z.P.S. and H.L. reviewed the original manuscript.

## Competing interests
The authors declare no competing interests.
