## [Peer Review File · Nature Communications]

PolyMOF nanoparticles constructed from intrinsically microporous polymer ligand towards scalable composite membranes for CO₂ separationREVIEWER COMMENTS

Reviewer #1 (Remarks to the Author):

The author presents a novel polymer-MOF system using a carboxylated polymer derived from intrinsic microporosity PIM-1. The polyMOFs exhibit enhanced ultramicroporosity and improved dispersion. It is an interesting work and I recommend its publication after addressing the following comments.

1. The author successfully synthesized crystalline polyMOFs by carefully controlling the ratio between the BDC and cPIM-1. I would like to know the number of repeating units in cPIM-1 and whether the molecular weight of cPIM-1 influences the structure of the synthesized polyMOF.
2. In Fig. 2(d), it is evident that the presence of the BDC ligand, regardless of the BDC to cPIM-1 ratio, leads to XRD peaks that closely resemble those of UiO-66. This suggests that Zr exhibits a higher tendency to coordinate with BDC, while cPIM-1 does not coordinate with the metal clusters. The FTIR spectra in Fig. 2(f) also support this observation, as the pattern of 0:1 is consistent with that of cPIM-1, while 4:1 is consistent with UiO-66.
3. In Fig. 2(e), it would be helpful if the author could provide clarification on how they determined whether the -COO- or COOH groups belong to BDC or cPIM-1.
4. The cPIM-1 ligand demonstrates the ability to interpenetrate through the MOF lattice by coordinating with metal oxo clusters, rather than simple ionic crosslinking. It would greatly enhance the study if the author could provide TEM images with nanoscale resolution synthesized with different BDC/cPIM-1 molar ratios, particularly focusing on the body phase and edge positions of the crystals. With an increased cPIM-1 molar ratio, the coordination probability is expected to increase significantly, potentially leading to noticeable effects on the structure. Additionally, it would be beneficial to report the yield of MOF crystals synthesized with different BDC/cPIM-1 molar ratios.
5. The caption of Fig. S13 appears to be inconsistent with the main text. It is suggested that the author correct it to "MIL-101" instead of "MOF-5".
6. Is there any reliable method to determine the location of cPIM-1 within the crystal structure?
7. While the author provided a clear BDC/cPIM-1 ligand molar ratio, it would be informative to include the molar ratio of the two ligands in the synthesized MOF structure.
8. In Fig. 7(a), the overall yellowish tone of the photo makes the border of the membrane unclear. Please consider enhancing the clarity of the image.
9. The membrane thickness in Fig. 7(c) may not be accurately represented since a small portion of the membrane surface is included. Additionally, the upper layer of the cross-section in Fig. 7(b) appears blurred, and a clearer image should be provided.
10. It is suggested to use the same membrane area for comparing the membrane performance in Fig. 7(e). Furthermore, the separation performance of the U20/PIM-1 TFC membrane should be included for a comprehensive analysis.

Reviewer #2 (Remarks to the Author):

I have carefully read the manuscript "PolyMOF nanoparticles constructed from intrinsically microporous polymer ligand towards scalable composite membranes for CO₂ separation". This manuscript proposes a multifunctional polyMOF system constructed from a microporous polymer ligand, which both modulates the characteristics of PolyMOF nanoparticles by a one-step synthesis and provides the framework with angstrom-scale microporosity for molecular sieving. The authors investigated the preparation process of PolyMOF nanoparticles and the gas separation properties of the composite membranes. The coordination reactions between PIM ligands and metal ions were investigated by detailed microscopy, spectroscopy and thermal analysis, which confirmed the successful synthesis of PIM-based PolyMOF nanoparticles. The results of membrane separation of gas mixtures show that the prepared PolyMOF nanoparticles have excellent dispersion and their molecular sieving ability can effectively improve the overall separation performance of the membranes, which provides a new idea for better control of the synthesis of MOF nanomaterials for the preparation of MMMs. Therefore, the manuscript can be considered by the journal, however, several issues should be further addressed.

(1) As mentioned in the manuscript, dynamic light scattering (DLS) demonstrates the excellent colloidal stability of PolyMOF with an average diameter of ~100 nm in DMF and THF. It should be noted that the DLS results only demonstrate to some extent that the nanoparticles are homogeneously dispersed in solution, but not their dispersion stability. The investigation of the dispersion stability of nanofillers in dispersions needs to be complemented by the characterization of the zeta potential.

(2) The authors analyzed the CO₂ and N₂ transport properties of PIM-1, U20/PIM-1, and pU20/PIM-1 membranes based on the solution-diffusion model in order to better investigate the gas transport mechanisms of MMMs, and obtained some reasonable results. However, from the whole paper, the improvement of membrane performance was mainly attributed to the contribution of PolyMOF nanoparticles, so additional analysis of the "contribution of nanofillers to the overall permeation selectivity of membranes" should be considered with reference to related literature.

(3) As stated by the authors in the manuscript, the PolyMOF prepared in this work can effectively enhance the separation performance of the membrane for CO₂ gas mixtures, as demonstrated by the actual test results. It should be noted that the long-term stable operation of the membrane is particularly important to achieve efficient separation of CO₂ gas mixtures, as it is related to the economic feasibility and practical application value of the whole membrane system. Although PIM is an important membrane material for gas separation, it has long been plagued by problems such as physical aging and plasticization. The authors mentioned in the manuscript that the PolyMOF prepared in this work can limit the movement of PIM polymer molecular chains to a certain extent and improve the resistance to plasticization of MMMs, but there seems to be no relevant research and discussion on stability aspect in the paper, which should be added.

(4) The authors should provide the full name of the chemicals for the first time and may use abbreviations in the future to avoid repetition of the full name.

(5) It is recommended to adjust the scale in Figure 5(d) and (f), and one can try to use nanometer as the scale unit.

(6) The entire manuscript is too long, so consider putting some of the content into a Supplementary Information document.

(7) The format of some references may not be standardized, please refer to the submission guidelines for revision.

(8) Language and expression should be further improved.

Reviewer #3 (Remarks to the Author):

The paper described the use of a synthetic route to more closely disperse MOF particles into a high permeance PIM-1 material. The results are enhanced porosity, enhanced mechanical properties and enhanced gas separation performances.

The paper is itself well written and the materials well characterized. However it ends up being a mixture of applied and fundamental without either end of the spectrum being fully developed.

If the authors had focused on the fundamental, then they should have demonstrated generality by demonstrating the principle within a number of matrix polymers besides PIM-1.

They instead also strove to demonstrate the general applicability to carbon capture and the separation of CO₂ and N₂. They did so by demonstrating relatively large area coatings, which is more of a technological feat, but hardly an interesting one. Moreover their materials, have at best a limited CO₂/N₂ selectivity of approximately 20, and this result is reported within a thick 3 micron film.

What concerns me is that they could easily have made thinner films, with much higher permeances, but chose not to. I suspect this is because the resulting films have an extreme aging behavior.

In their report, they show in Table S6, that there is a dramatic loss of permeance within 14 days. Although the use of the PolyMoF system seems to slow down the aging, there is not enough data to show how far the aging will proceed within a longer time frame (e.g. 6-12 months)

Whilst the authors have openly accepted the limitations of using a well known polymer that ages (i.e. PIM-1), they would have been better to demonstrate the phenomenon within a non-aging polymer matrix.

Another aspect is that although they claim that selectivity of 20 is fine for CO₂/N₂ separation, it is not (in my opinion), even via the simulations in the paper they have cited by Merkel et al. Ideal gas selectivity of 50 are required for a membrane measured in its membrane state, since there are almost always losses in selectivity, as a result of the modulation process (spiral or plate and frame).

In summary, I think this is a valuable incremental work, which would be an excellent addition to a specialized journal, but lacks the degree of novelty (since functionalizing MOFs to enhance dispersability is a known approach), and lacks the impact of high separation performances and long term stability to be attractive to the application for carbon capture.

Reply to Reviewers' comments (*Manuscript number: NCOMMS-23-23454*)

-Reviewer 1:

The author presents a novel polymer-MOF system using a carboxylated polymer derived from intrinsic microporosity PIM-1. The polyMOFs exhibit enhanced ultramicroporosity and improved dispersion. It is an interesting work and I recommend its publication after addressing the following comments.

Response:

We appreciate your understanding of the value of this work. In this response letter, we tried to address all the comments from the reviewers as much as possible.

1. The author successfully synthesized crystalline polyMOFs by carefully controlling the ratio between the BDC and cPIM-1. I would like to know the number of repeating units in cPIM-1 and whether the molecular weight of cPIM-1 influences the structure of the synthesized polyMOF.

Response:

We appreciate the reviewer's comment. We tried measuring the molecular weight of cPIM-1 using the gel permeation chromatography (GPC) method. Unfortunately, we always got unacceptable results (low M_w values that were not consistent with being able to make freestanding cPIM-1 films). Presumably, the polymer may get stuck to the GPC walls in the process, and only the low M_w elements get through. We believe that this is why the previous study also did not report the molecular weight of cPIM-1 (Macromolecules 2020, 53, 6220–6234). Nevertheless, we expect that the molecular weight of cPIM-1 would affect the morphological and structural properties of the resultant polyMOFs (Chem. Commun., 53, 3058–3061 (2017)).

2. In Fig. 2(d), it is evident that the presence of the BDC ligand, regardless of the BDC to cPIM-1 ratio, leads to XRD peaks that closely resemble those of UiO-66. It suggests that Zr tends to coordinate with BDC, while cPIM-1 does not coordinate with the metal clusters. The FTIR spectra in Fig. 2(f) also support this observation, as the pattern of 0:1 is consistent with that of cPIM-1, while 4:1 is consistent with UiO-66.

Response:

We appreciate the reviewer's comment. The XRD results suggest that crystalline polyMOFs could be synthesized, but these do not provide information on the coordination chemistry. As shown in the Main text (Fig. 2), although polyUiO-66(0:1) (which does not include BDC) exhibited a similar FT-IR spectrum compared with that of cPIM-1, it still shows a significant presence of coordinated COO^- groups as evidenced by the FT-IR peak at 1583 cm^{-1} and ssNMR spectrum ($-\text{COO}^-$, at ~ 171 ppm). In addition, the residual mass at 800°C obtained from TGA curves under air purge (18.3 wt.%) is far above that typically observed in other studies (< 5 wt.%) on ionic crosslinking of cPIM-1 (Fig. S9 and Table S2), even for polyUiO-66(0:1). These results confirm the coordination between Zr oxo clusters and cPIM-1 ligands.

3. In Fig. 2(e), it would be helpful if the author could provide clarification on how they determined

whether the -COO- or COOH groups belong to BDC or cPIM-1.

Response:

We appreciate the reviewer's suggestion. As shown in Fig. 2e, in contrast to the cPIM-1 that showed only a -COOH peak, the NMR spectra of polyUiO-66(0:1) (which does not include BDC) exhibited both -COOH and -COO⁻ portions. It suggests specific coordinations between Zr metal and cPIM-1 ligands exist in all polyUiO-66 samples. FT-IR and TGA analyses further proved the coordination between Zr oxo clusters and cPIM-1 ligands in polyUiO-66 samples. In the revised manuscript, we added references to clarify the position of both -COOH and -COO⁻ peaks in NMR spectra.

Changes made:

- The following sentence was changed in the revised manuscript.

Original:

“As the cPIM-1 concentration for the polyUiO-66 synthesis increased, the peak for uncoordinated groups (-COOH, at ~163 ppm) became more intense, while that for coordinated groups (-COO⁻, at ~171 ppm) became broader and moved toward the peak for the -COOH group (Fig. 2e).”

Revised:

“As the cPIM-1 concentration for the polyUiO-66 synthesis increased, the peak for uncoordinated groups (-COOH, at ~163 ppm)⁴⁴ became more intense, while that for coordinated groups (-COO⁻, at ~171 ppm)³⁹ became broader and moved toward the peak for the -COOH group (Fig. 2e).”

Ref. 39: J. Am. Chem. Soc. 142, 10863–10868 (2020)

Ref. 44: Macromolecules 53, 6220–6234 (2020)

: Main text, Results, page 8

4. The cPIM-1 ligand can interpenetrate through the MOF lattice by coordinating with metal oxo clusters rather than simple ionic crosslinking. It would significantly enhance the study if the author could provide TEM images with nanoscale resolution synthesized with different BDC/cPIM-1 molar ratios, mainly focusing on the body phase and edge positions of the crystals. With an increased cPIM-1 molar ratio, the coordination probability is expected to increase significantly, potentially leading to noticeable effects on the structure. Additionally, it would be beneficial to report the yield of MOF crystals synthesized with different BDC/cPIM-1 molar ratios.

Response:

We appreciate the reviewer's suggestions. In the revised manuscript, we added the high-resolution TEM images of the polyUiO-66 nanoparticles with different BDC/cPIM-1 molar ratios and the corresponding comments. The yield of polyUiO-66 nanoparticles was also added in the experimental section.

Changes made:

- The following figure was added to the revised manuscript.

Fig. S7. High-resolution TEM images of (a-b) UiO-66, (c-d) polyUiO-66(8:1), (e-f) polyUiO-66(4:1), and (g-h) polyUiO-66(2:1).

: Supplementary Information

- The following sentence was added to the revised manuscript.

“High-resolution TEM images also reveal that the morphology of the polyUiO-66 nanoparticles becomes rougher by increasing the cPIM-1 concentration (Fig. S7).”

: Main text, Results, page 7

- The following sentence was added to the revised manuscript.

“The yield of polyMOF nanoparticles was approximately 60–70% by weight.”

: Main text, Methods, page 27

5. The caption of Fig. S13 appears to be inconsistent with the main text. It is suggested that the author correct it to “MIL-101” instead of “MOF-5”.

Response:

We greatly appreciate the reviewer’s comment. We corrected the typo in the revised manuscript.

Changes made:

- The following sentence was changed in the revised manuscript.

Original:

Fig. S13. SEM images of (a) MOF-5, (b) polyMOF-5(4:1), and (c) polyMOF-5(0:1) powder (scale bar = 200 nm).

Revised:

Fig. S14. SEM images of (a) MIL-101, (b) polyMIL-101(4:1), and (c) polyMIL-101(0:1) powder (scale bar = 200 nm).

: Supplementary Information

6. *Is there any reliable method to determine the location of cPIM-1 within the crystal structure?*

Response:

We appreciate the reviewer's comment. A previous study has suggested a method to reveal the structure of the polymer backbones in polyMOF materials by combining 2D NMR and molecular simulations (J. Am. Chem. Soc. 142, 10863–10868). We are also interested in such a detailed analysis of our polyMOF materials, which will be discussed in a separate paper since the current work focuses on the first development of the PIM-based polyMOF materials and their applications for CO₂ separation.

7. *While the author provided a clear BDC/cPIM-1 ligand molar ratio, it would be informative to include the molar ratio of the two ligands in the synthesized MOF structure.*

Response:

We appreciate the reviewer's comment. The formation of the linker or cluster defects in the resultant polyMOFs is inevitable, particularly for the polyUiO-66 samples, since we use formic acid as a modulator (Chem. Mater. 285, 3749–3761). This makes it very difficult to determine the precise ratio of the two ligands. Alternatively, we could calculate the concentration of the cPIM-1 ligand incorporated into the synthesized polyMOFs by TGA data (Table S2 and S3).

8. *In Fig. 7(a), the overall yellowish tone of the photo makes the border of the membrane unclear. Please consider enhancing the clarity of the image.*

Response:

We appreciate the reviewer's suggestion. In the revised manuscript, we tuned the contrast and brightness of the photo images to enhance the clarity.

Changes made:

- The following figure was added to the revised manuscript.

Fig. 7. (a) Photo images of as-prepared pU20/PIM-1 TFN membranes depending on membrane area.

: Main text, Results, page 24

9. The membrane thickness in Fig. 7(c) may not be accurately represented since a small portion of the membrane surface is included. Additionally, the upper layer of the cross-section in Fig. 7(b) appears blurred, and a clearer image should be provided.

Response:

We fully agree with the reviewer's comment. First, we double-checked the SEM image of pU20/PIM-1 TFN membrane (Fig. R1), which showed almost identical membrane thickness (2.6 μm) compared to the value reported in the original manuscript (2.7 μm). This may ensure the uniformity of the membrane thickness of the pU20/PIM-1 TFN membrane.

Fig. R1. Cross-sectional SEM images of pU20/PIM-1 TFN membranes.

Second, we also double-checked the SEM image of the PIM-1 TFC membrane to get a clearer image (please see below), which showed almost identical membrane thickness (2.6 μm) compared to the value reported in the original manuscript (2.5 μm). We replaced the original image with the new one in the revised manuscript.

Changes made:

- The following figure was added to the revised manuscript.

Fig. 7. (b) Cross-sectional SEM images of (b) PIM-1 TFC membrane ($3 \times 3 \text{ cm}^2$) (scale bar = $2 \mu\text{m}$).

: Main text, Results, page 24

- The following sentence was changed in the revised manuscript.

Original: A $3 \times 3\text{-cm}^2$ -sized TFC membrane consisting of $\sim 2.5 \mu\text{m}$ -thick pure PIM-1 as a selective layer was also prepared (Fig. 7b) as a control sample,

Revised: A $3 \times 3\text{-cm}^2$ -sized TFC membrane consisting of $\sim 2.6 \mu\text{m}$ -thick pure PIM-1 as a selective layer was also prepared (Fig. 7b) as a control sample,

: Main text, Results, page 22

10. It is suggested to use the same membrane area for comparing the membrane performance in Fig. 7(e). Furthermore, the separation performance of the U20/PIM-1 TFC membrane should be included for a comprehensive analysis.

Response:

We appreciate the reviewer's suggestions. We plotted the $20 \times 20 \text{ cm}^2$ -sized pU20/PIM-1 TFN data in Fig. 7e to emphasize the scalability of this membrane since it showed almost identical CO_2/N_2 separation performances by varying the membrane area (Fig. 7d). Also, in the revised manuscript, we added the separation performance of the U20/PIM-1 TFN and the corresponding comments.

Changes made:

- The following figure was added to the revised manuscript.

Fig. S25. Cross-sectional SEM image of U20/PIM-1 TFN membranes ($3 \times 3 \text{ cm}^2$) (scale bar = 2 μm).

: Supplementary Information

- The following sentence was added to the revised manuscript.

“In contrast, ~3.3 μm-thick U20/PIM-1 TFC membrane showed a similar CO₂ permeance (~2400 GPU) with that of PIM-1 TFC membrane while accompanying a significantly reduced CO₂/N₂ selectivity (10.5), which is attributed to the severe agglomeration of UiO-66 nanoparticles (Fig. S25).”

: Main text, Results, page 22

-Reviewer 2:

I have carefully read the manuscript “PolyMOF nanoparticles constructed from intrinsically microporous polymer ligand towards scalable composite membranes for CO₂ separation”. This manuscript proposes a multifunctional polyMOF system constructed from a microporous polymer ligand, which both modulates the characteristics of PolyMOF nanoparticles by a one-step synthesis and provides the framework with angstrom-scale microporosity for molecular sieving. The authors investigated the preparation process of PolyMOF nanoparticles and the gas separation properties of the composite membranes. The coordination reactions between PIM ligands and metal ions were investigated by detailed microscopy, spectroscopy and thermal analysis, which confirmed the successful synthesis of PIM-based PolyMOF nanoparticles. The results of membrane separation of gas mixtures show that the prepared PolyMOF nanoparticles have excellent dispersion and their molecular sieving ability can effectively improve the overall separation performance of the membranes, which provides a new idea for better control of the synthesis of MOF nanomaterials for the preparation of MMMs. Therefore, the manuscript can be considered by the journal, however, several issues should be further addressed.

Response:

We appreciate your understanding of the value of this work. In this response letter, we tried to address all the comments from the reviewers as much as possible.

(1) As mentioned in the manuscript, dynamic light scattering (DLS) demonstrates the excellent

colloidal stability of PolyMOF with an average diameter of ~100 nm in DMF and THF. It should be noted that the DLS results only demonstrate to some extent that the nanoparticles are homogeneously dispersed in solution, but not their dispersion stability. The investigation of the dispersion stability of nanofillers in dispersions needs to be complemented by the characterization of the zeta potential.

Response:

We appreciate the reviewer's comment. When water was used as the solvent, the zeta potential of UiO-66 and polyUiO-66(4:1) dispersion was +37.7 mV and -26.3 mV, respectively. Of note, the negative charge of polyUiO-66(4:1) may be attributed to the uncoordinated carboxylic groups of cPIM-1 ligands. The higher absolute value of the zeta potential of UiO-66 than that of polyUiO-66(4:1) indicates its better colloidal stability in an aqueous solution, which is opposite to our results. However, this is not the case for the organic solvents, particularly for the low dielectric constant solvents (*i.e.*, THF), since ionization of the particles is only partially possible. Indeed, we could not obtain an acceptable zeta potential for each sample since the analysis always showed very low potential (± 1 mV) with very high standard deviations. Thus, aside from the surface charge, we conclude that the existence of unoccupied cPIM-1 ligands on their external surface may enhance colloidal stability based on the 'like dissolves like' rule, especially in good solvents for cPIM-1 such as THF and DMF.

(2) The authors analyzed the CO₂ and N₂ transport properties of PIM-1, U20/PIM-1, and pU20/PIM-1 membranes based on the solution-diffusion model to better investigate the gas transport mechanisms of MMMs, and obtained some reasonable results. However, from the whole paper, the improvement of membrane performance was mainly attributed to the contribution of PolyMOF nanoparticles, so additional analysis of the "contribution of nanofillers to the overall permeation selectivity of membranes" should be considered with reference to related literature.

Response:

We appreciate the reviewer's comment. To examine the CO₂/N₂ separation abilities of polyUiO-66(4:1) itself, in the original manuscript, we measured the CO₂/N₂ adsorption selectivity of the polyUiO-66(4:1) nanoparticles. The CO₂/N₂ adsorption selectivity of polyUiO-66(4:1) was improved compared to control UiO-66 by 27% at 0.1 bar and 14% at 1 bar. Notably, the obtained CO₂/N₂ selectivity of polyUiO-66(4:1) is highest among the UiO-66-based adsorbents with a similar level of CO₂ uptake (Table S5). The excellent CO₂/N₂ selectivity of polyUiO-66(4:1) is mainly attributed to the presence of ultramicropores, especially in the 3–4 Å range, which may contribute to the more pronounced molecular sieving effect that allows the diffusion of smaller CO₂ molecules while retarding that of larger N₂ molecules. As a result, the pU20/PIM-1 MMMs exhibited both enhanced CO₂ permeability and CO₂/N₂ selectivity compared to the control PIM-1 membrane. In the revised manuscript, we added relevant references that describe a similar context to our explanation.

Changes made:

- The following sentence was changed in the revised manuscript.

Original:

These features are ascribed to the effective molecular sieving by ultramicroporous polyUiO-66(4:1) filler, uniform particle dispersion, and improved filler–matrix compatibility, which are responsible for the simultaneous increase in CO₂ permeability and CO₂/N₂ selectivity of pU20/PIM-1.

Revised:

These features are ascribed to the effective molecular sieving by ultramicroporous polyUiO-66(4:1) filler, uniform particle dispersion, and improved filler–matrix compatibility, which are responsible for the simultaneous increase in CO₂ permeability and CO₂/N₂ selectivity of pU20/PIM-1^{65,66}.

Ref. 65: Science 378, 1189–1194 (2022)

Ref. 66: Science 376, 1080–1087 (2022)

: Main text, Results, page 19

(3) As stated by the authors in the manuscript, the PolyMOF prepared in this work can effectively enhance the separation performance of the membrane for CO₂ gas mixtures, as demonstrated by the actual test results. It should be noted that the long-term stable operation of the membrane is particularly important to achieve efficient separation of CO₂ gas mixtures, as it is related to the economic feasibility and practical application value of the whole membrane system. Although PIM is an important membrane material for gas separation, it has long been plagued by problems such as physical aging and plasticization. The authors mentioned in the manuscript that the PolyMOF prepared in this work can limit the movement of PIM polymer molecular chains to a certain extent and improve the resistance to plasticization of MMMs. Still, there seems to be no relevant research and discussion on the stability aspect in the paper, which should be added.

Response:

We greatly appreciate the reviewer’s insightful comments. As stated in the original manuscript, although significant aging-induced permeance reduction was found in both PIM-1 TFC and pU20/PIM-1 TFN membranes, the extent was much less for the TFN membrane, possibly due to the favorable filler–matrix interactions. In addition, the aging behavior can be addressed by nonsolvent-based rejuvenation, or we could exploit the aging-induced selectivity enhancement. To demonstrate the potential of these post-treatments, we further tested the long-term stability of the PIM-1 TFC and pU20/PIM-1 TFN membranes with methanol-assisted rejuvenation of the 14-days-aged membranes by following the previous report (J. Membr. Sci. 520, 671–678 (2016)). The results demonstrate the methanol post-treatments could rejuvenate the aged membranes, which was also repeatable. We anticipate that this simple post-treatment would also be viable in the real process if a proper condition is optimized (e.g., methanol vapor treatment (ACS Macro Lett. 12, 113–117 (2023))). We added the long-term stability results and the corresponding comments in the revised manuscript.

Changes made:

- The following figure was added to the revised manuscript.

Fig. S26. Long-term CO₂/N₂ separation performances of PIM-1 TFC and pU20/PIM-1 TFN membranes with methanol-assisted rejuvenation of the 14-day-aged membranes. For rejuvenation, the aged membranes were soaked in methanol for 24 h and dried in a vacuum oven at room temperature for 24 h.

: Supplementary Information

- The following sentence was added to the revised manuscript.

“For example, the 14-days-aged PIM-1 TFC and pU20/PIM-1 TFN membranes could be rejuvenated by simply soaking them into methanol, which was also repeatable (Fig. S26).”

: Main text, Results, page 23

- Table S6 in the original manuscript was removed to avoid the overlap with Fig. S26 in the revised manuscript.

: Supplementary Information

(4) The authors should provide the full name of the chemicals for the first time and may use abbreviations in the future to avoid repetition of the full name.

Response:

We appreciate the reviewer’s suggestion. In the revised manuscript, we double-checked the names and abbreviations of the chemicals and samples.

(5) It is recommended to adjust the scale in Figure 5(d) and (f), and one can try to use nanometer as the scale unit.

Response:

We appreciate the reviewer's suggestion. We changed the scale unit in Fig. 5d and f in the revised manuscript.

Changes made:

- The following figure was added to the revised manuscript.

Fig. 5. Cross-sectional SEM images of (c-d) U20/PIM-1 and (e-f) pU20/PIM-1 membranes.
: Main text, Results, page 18

(6) The entire manuscript is too long, so consider putting some of the content into a Supplementary Information document.

Response:

We appreciate the reviewer's suggestion. To shorten the manuscript, we moved the paragraph on the cPIM-1 synthesis to the Supplementary Information.

(7) The format of some references may not be standardized. Please refer to the submission guidelines for revision.

Response:

We appreciate the reviewer's suggestion. We double-checked the reference style in the revised manuscript according to the submission guidelines.

(8) Language and expression should be further improved.

Response:

We appreciate the reviewer's suggestion. We double-checked all the grammatical errors and typos in the revised manuscript.

-Reviewer 3:

The paper described using a synthetic route to more closely disperse MOF particles into a high permeance PIM-1 material. The results are enhanced porosity, enhanced mechanical properties, and enhanced gas separation performances.

The paper is itself well written and the materials well characterized. However it ends up being a mixture of applied and fundamental without either end of the spectrum being fully developed.

If the authors had focused on the fundamental, then they should have demonstrated generality by demonstrating the principle within a number of matrix polymers besides PIM-1.

They instead also strove to demonstrate the general applicability to carbon capture and the separation of CO₂ and N₂. They did so by demonstrating relatively large area coatings, which is more of a technological feat, but hardly an interesting one. Moreover their materials, have at best a limited CO₂/N₂ selectivity of approximately 20, and this result is reported within a thick 3 micron film.

What concerns me is that they could easily have made thinner films, with much higher permeances, but chose not to. I suspect this is because the resulting films have an extreme aging behavior.

In their report, they show in Table S6, that there is a dramatic loss of permeance within 14 days. Although the use of the PolyMoF system seems to slow down the aging, there is not enough data to show how far the aging will proceed within a longer time frame (e.g. 6-12 months)

Whilst the authors have openly accepted the limitations of using a well known polymer that ages (i.e. PIM-1), they would have been better to demonstrate the phenomenon within a non-aging polymer matrix.

Another aspect is that although they claim that selectivity of 20 is fine for CO₂/N₂ separation, it is not (in my opinion), even via the simulations in the paper they have cited by Merkel et al. Ideal gas selectivity of 50 are required for a membrane measured in its membrane state, since there are almost always losses in selectivity, as a result of the modulation process (spiral or plate and frame).

In summary, I think this is a valuable incremental work, which would be an excellent addition to a specialized journal, but lacks the degree of novelty (since functionalizing MOFs to enhance dispersability is a known approach), and lacks the impact of high separation performances and long term stability to be attractive to the application for carbon capture.

Response:

We greatly appreciate the reviewer's insightful comments. We admit that the original manuscript

did not provide sufficient results on the concerns of the reviewers. In this response letter, we tried to address all the comments from the reviewers as much as possible.

First, we prepared the MMMs containing 20 wt.% of UiO-66 and polyUiO-66(4:1) nanoparticles with different polymer matrices such as 6FDA-DAM, 6FDA-DAM:DABA(3:2), and Matrimid. As with the PIM-1 case, we could observe the improved CO₂ separation performances in the MMMs containing polyUiO-66(4:1) filler with different polymer matrices. This emphasizes the versatility of the PIM-based polyMOF filler design. Of note, the extent of enhancement effect in CO₂ separation performance varies depending on the polymer matrix, potentially due to the different polymer–filler interfacial compatibilities as observed in the previous study (Nat. Commun. (2023) DOI: 10.1038/s41467-023-37479-9). In the revised manuscript, we added these data and the corresponding comments.

Changes made:

- The following figures were added to the revised manuscript.

Fig. S22. XRD spectra of (a) 6FDA-DAM, (b) 6FDA-DAM:DABA(3:2), and (c) Matrimid MMMs containing 20 wt.% of UiO-66 (U20) or polyUiO-66(4:1) nanoparticles, respectively.

Fig. S23. Pure-gas separation performances of pure polymers and MMMs plotted with 2008 Robeson upper bound for (a) CO₂/N₂ and (b) CO₂/CH₄ separation.

: Supplementary Information

- The following sentences were added to the revised manuscript.

“The improved CO₂ separation performances were also observed in the MMMs containing polyUiO-66(4:1) filler with different polymer matrices (Fig. S22 and S23). This emphasizes the versatility of the PIM-based polyMOF filler design.”

: Main text, Results, page 19

“6FDA-DAM and 6FDA-DAM:DABA(3:2) polymers were purchased from Akron Polymer Systems (USA). Matrimid® 5218 polymer was purchased from Alfa Aesar (USA).”

: Main text, Methods, page 25–26

Second, we fully understand the reviewer’s concern with the aging issue in the PIM-1-based TFC membranes. We could prepare a thinner membrane to boost the CO₂ permeance. Still, we failed to get acceptable long-term stability due to the intensified aging behavior, as the reviewer pointed out. To address this issue, we tested the long-term stability of the PIM-1 TFC (2.6 μm) and pU20/PIM-1 TFN (2.7 μm) membranes with methanol-assisted rejuvenation of the 14-days-aged membranes by following the previous report (J. Membr. Sci. 520, 671–678 (2016)). The results demonstrate the methanol post-treatments could rejuvenate the aged membranes, which was also repeatable. We anticipate that this simple post-treatment would also be viable in the real process if a proper condition is optimized (e.g., methanol vapor treatment (ACS Macro Lett. 12, 113–117 (2023))). In the revised manuscript, we added the long-term stability results and the corresponding comments – Please refer to the response to reviewer #2’s comment (3) above.

We also acknowledge that non-aging polymer materials such as Pebax would be more beneficial regarding long-term stability. However, the main purpose of this study is to demonstrate the

concept that it is effective to prepare high-performance MMMs by incorporating polyMOF materials into the polymer matrix that shares a similar structure with the polymer ligand, which improves the filler–matrix interfacial compatibility based on ‘like dissolves like’ rule. That’s the main reason why we chose PIM-1 as a polymer matrix, which demonstrated the significant enhancement in both CO₂ permeability and CO₂/N₂ (CO₂/CH₄) selectivity simultaneously. On the other hand, the developed polyUiO-66 nanoparticles showed poor colloidal stability in water and alcohols, which makes it difficult to prepare Pebax MMMs containing polyMOFs. In addition, Pebax materials generally exhibit a low CO₂ permeability due to crystalline PEO blocks. This forces us to prepare ultrathin (<50 nm) membranes to satisfy the CO₂ permeance requirement (>1000 GPU), which is very susceptible to defect formations that eventually deteriorate CO₂/gas selectivities.

Lastly, we admit that the CO₂/N₂ selectivity of 20–30 of the developed pU20/PIM-1 material may not be enough for modulization. However, as mentioned above, the main purpose of this study is to demonstrate the concept of polyMOF materials containing cPIM-1 ligands and their potential applications as a filler to improve both CO₂ permeability and CO₂/gas selectivity of MMMs. As stated in the original manuscript, we envision that it is possible to find “sweet spots” by utilizing the aging-induced selectivity enhancement effects as demonstrated in our previous work (*J. Membr. Sci.* 672, 121438 (2023)). For example, during this revision, we found that a 14-day-aged pU20/PIM-1 TFN membrane exhibited CO₂/N₂ selectivity of >30 while the decrease in CO₂ permeance almost plateaued. Also, we could further consider optimizing the coating conditions and post-treatment steps to improve the CO₂/N₂ selectivity of pU20/PIM-1, which is intensively ongoing in our group and will be reported in a separate paper.

REVIEWERS' COMMENTS

Reviewer #1 (Remarks to the Author):

The authors have addressed the concerns and comments. The revised manuscript is suggested to be published in Nature Communications.

Reviewer #2 (Remarks to the Author):

The authors have provided detailed explanations and revisions to the issues we raised, and therefore we believe the work is ready for publication in this journal.

Reviewer #3 (Remarks to the Author):

In my original review, I questioned the issues of addressing aging, especially in thin films. I also questioned why other materials had not been explored as matrices for the PolyMOF additives.

In response, the reviewers have adequately acknowledged that although their paper does not tackle aging in PIM-1, this is not the central theme of their paper.

In addition, they have made efforts to demonstrate the addition of the PolyMOF in other systems besides PIM-1.

Therefore I have no further objections to the publication of this paper.